



# iPyCLES v1.0: A New Isotope-Enabled Large-Eddy Simulator for Mixed-Phase Clouds

Zizhan Hu[1], Yiran Peng[1], Mengke Zhu[1], and Jonathon S. Wright[1]

[1]Ministry of Education Key Laboratory for Earth System Modeling, Department of Earth System Science, Tsinghua University, Beijing, China

**Correspondence:** Jonathon S. Wright (jswright@tsinghua.edu.cn)

**Abstract.** Recent field campaigns and advances in observational techniques have yielded a wealth of observations of stable water isotopes in the atmosphere, but the potential for these observations to constrain parameterized physics in global models has yet to be fully realized. Here, we introduce the isotope-enabled Python Cloud Large-Eddy Simulation model (iPyCLES) for mixed-phase clouds. Isotopic tracers are implemented in a parallel passive water cycle and experience all processes and phase
changes that affect the model's prognostic total water variable. Isotopic fractionation occurs during cloud and precipitation processes as well as surface evaporation, with facilities for applying external forcing. In addition to isotopic tracers, we extend the two-moment warm cloud microphysics scheme to enable prognostic simulation of cloud liquid water and ice while eliminating dependence on saturation adjustment. Relative to a one-moment mixed-phase scheme with saturation adjustment, the new microphysical scheme yields substantial benefits in simulating phase partitioning and isotopic exchange in mixed-phase
regions. The LES model is based on an energetically-consistent implementation of the anelastic equations and employs high-order weighted, essentially non-oscillatory numerics, and is therefore theoretically suitable for simulations spanning the gray zone of the convective spectrum. In this initial evaluation, we present the results of test cases for non-precipitating subtropical shallow cumulus, precipitating subtropical shallow cumulus, and precipitating Arctic mixed-phase stratocumulus clouds. The iPyCLES simulations agree well with available observations and previous model simulations in all three cases, with distinct
signatures among the cases that highlight the added potential of isotopic tracers. The benefits of the revised microphysical scheme are especially evident in the Arctic mixed-phase cloud test case, with vapor-liquid-ice exchange within the cloud producing a conspicuous peak in deuterium excess near the top of the cloud. As an idealized testbed, the iPyCLES model can bridge gaps between cloud chamber experiments, real-world observations, and global and regional models, allowing information provided by water isotopes to be translated more effectively into observational constraints for cloud and boundary layer
parameterizations.

## 1 Introduction

Despite considerable progress, clouds remain a large source of uncertainties in projections of global and regional climate change (Arias et al., 2021). Through their roles in modulating the distributions of precipitation and latent heating and interactions with both shortwave and longwave radiation, changes in the formation and distribution of clouds have significant impacts



on both the energy and water cycles. It is therefore critical to improve our understanding and ability to simulate the microphysical processes involved in the evolution of clouds and precipitation (Waliser et al., 2011; Li et al., 2012; Michibata et al., 2019; Schneider et al., 2019; Hofer et al., 2019).

Cloud microphysical parameterizations are used in atmospheric models to represent the bulk behaviors of small droplets and particles in clouds and their conversion to rain and snow (Gettelman et al., 2012; Khain et al., 2015; Morrison and Pinto, 2006;

Stein et al., 2015; Wu and Petty, 2010). Owing to the complexity of these behaviors, parameterizations are often based on statistical approaches that combine theoretical expectations with available measurements (Tapiador et al., 2019; Phillips et al., 2008). The reliability of microphysical parameterizations is thus constrained by both limitations in our understanding and a lack of high-quality measurements, especially with respect to atmospheric ice particles and the effects of aerosols on cloud formation (Morrison et al., 2020; Zhao et al., 2017; Kreidenweis et al., 2019). The understanding of cloud microphysical processes has

vastly improved since 1960 with advances in observational techniques, including laboratory experiments conducted under idealized conditions, data collected during field expeditions, and cloud properties observed by satellite instruments (Bailey and Hallett, 2004; McMurry, 2000; Rasmussen, 1995; McFarquhar et al., 2007; Wang et al., 2016; Rosenfeld et al., 2014; Lamb et al., 2023). Nonetheless, significant gaps remain between cloud microphysics in the real world and cloud microphysics as represented by atmospheric models (Zhang and Li, 2013; Webb et al., 2017; Li et al., 2020). As outlined by Morrison et al.

(2020), at least two fundamental challenges remain to be addressed: (1) the "parameterization square" challenge, which arises from the coupling of microphysical parameterizations with parameterizations of unresolved cloud structure and dynamics; and (2) insufficient understanding of basic microphysical processes as revealed through observations, especially the formation and evolution of ice particles.

Large Eddy Simulation (LES) models with explicit representations of clouds and precipitation are an important tool for

addressing the "parameterization square" challenge (Wang et al., 2009; Morrison et al., 2020). Most mesoscale models operate at kilometer-scale resolutions, which fall within the gray zone where the grid spacing is comparable to the size of the most energetic eddies in moist convective systems (Wyngaard, 2004). Consequently, these models often show a strong resolution dependence of convective properties (Ching et al., 2014; Hanley et al., 2015) and a high susceptibility to details in the method used for turbulence closure (Bryan and Rotunno, 2009; Shi et al., 2019; Tompkins and Semie, 2017). By contrast, LES models

explicitly resolve many processes involved in cloud formation that must be parameterized in larger-scale models, including the vertical distribution of cloud cover and changes in cloud size over time. LES models can therefore be used to test microphysical parameterizations without the compounding effects of parameterized convection and boundary layer processes (Morrison et al., 2020; Stevens et al., 2020). LES models are designed to resolve much of the three-dimensional turbulence spectrum, especially the larger-scale turbulent motions that contain most of the kinetic energy, although smaller-scale turbulent motions are often

parameterized (Stevens et al., 2020; Dyroff et al., 2015). LES-based research over the past few decades includes studies focusing on dry convective boundary layers (Schmidt and Schumann, 1989; Sullivan and Patton, 2011), stably stratified boundary layers (Beare et al., 2006), shallow cumulus convection (Jiang and Cotton, 2000; Siebesma et al., 2003), stratocumulus-topped marine boundary layers (Kogan et al., 1995; van der Dussen et al., 2015), and mixed-phase clouds in polar regions (Ovchinnikov et al., 2014; Savre and Ekman, 2015; Zhang et al., 2020). Moreover, recent expansions of computational resources



support the expanded use of LES models for studying cumulus convection at resolutions and spatial scales that were previously unattainable (Lane and Sharman, 2014). Several recent studies have pushed boundaries in this direction to examine turbulent motions within deep convective clouds (Parodi and Tanelli, 2010; Machado and Chaboureau, 2015; Verrelle et al., 2015; Gu et al., 2020), including some extreme cumulonimbus cases (Khairoutdinov et al., 2009; Lane and Sharman, 2014; Heath et al., 2017).

Stable water isotopologues (hereafter "isotopes") offer the potential for significant progress on the second challenge, which concerns the difficulty of establishing observational constraints on microphysical processes. Isotopes have been widely used in atmospheric science as tracers for studying transport and physical processes in the atmosphere (Jouzel, 1986; Yoshimura, 2015; Galewsky et al., 2016; Wright et al., 2017). The hydrometeorological utility of stable water isotopes is rooted in the occurrence of isotopic fractionation during phase changes, as isotopically heavier molecules ($^1H^2H^{16}O$: 0.02%, $^1H_2^{18}O$: 0.20%) prefer-

entially enter or remain in the condensed-phase relative to the isotopically light molecule that comprises most of Earth's water ($^1H_2^{16}O$: 99.73%). Each detectable isotope has the potential to provide additional information about physical processes in the atmosphere that cannot be readily or directly observed, with a degree of independence that depends on the extent of differential fractionation among the isotopes. Isotopic tracers have been employed in hydrology and paleoclimatology since the 1950s, as reviewed by Dansgaard (1964) and Gat et al. (1996), among others. Recent advances in technologies for observing and

simulating atmospheric isotopes in the atmosphere have fueled a second wave of studies leveraging the isotopic compositions of atmospheric water vapor and precipitation for insights into the movement, mixing, and phase transitions of water in the atmosphere (Sturm et al., 2005; Smith et al., 2006; Worden et al., 2007; Lee et al., 2006, 2009, 2012b; Dyroff et al., 2015; Moore et al., 2016; Lamb et al., 2017; Schneider et al., 2017; Eckstein et al., 2018; Prasanna et al., 2018; Risi et al., 2020).

Isotopic fractionation stems from two primary characteristics of heavy isotopes relative to the standard water molecule (Rozan-

ski et al., 2001). First, heavier isotopes possess greater molecular mass and therefore diffuse more slowly than lighter isotopes. Second, heavy isotopes have greater binding energies, so that heavy isotopes are progressively more prevalent in the liquid and solid phases relative to the vapor phase (Rozanski et al., 2001; Coplen et al., 2000). In the atmosphere, these disparities collectively contribute to a preference for heavy isotopes to enter or remain in the condensed phase during phase changes, such as condensation and deposition. Fractionation during phase changes that occur in thermodynamic equilibrium is extremely

well described by temperature-dependent fractionation factors derived from laboratory experiments (Majoube, 1971; Rozanski et al., 2001). However, much of the potential for isotopes to provide constraints on physical processes in the atmosphere derives from departures from equilibrium fractionation, often referred to as "kinetic fractionation." Distinct isotopic signatures may therefore help to identify atmospheric processes that involve significant disequilibrium during phase transitions. Signatures of kinetic fractionation have been linked to turbulent evaporative fluxes in the boundary layer (Craig and Gordon, 1965; Feng

et al., 2019; Risi et al., 2019), evaporative loss from raindrops in unsaturated air (Worden et al., 2007; Lee and Fung, 2008), and the depth and phase partitioning of the mixed-phase layer in clouds (Bolot et al., 2013).

The development of a heirarchy of isotope-enabled atmospheric models (Hoffmann et al., 1998; Blossey et al., 2010; Pfahl et al., 2012; Bolot et al., 2013; Wei et al., 2018; Brady et al., 2019) has further enabled innovative applications of stable water isotopes to advance our understanding of the atmospheric water cycle, from rain recycling and mixing in the marine boundary





layer (Lee et al., 2012b; Risi et al., 2019), to the depths of the troposphere (Jouzel et al., 1980; Bolot et al., 2013; Eckstein et al., 2018) and even into the stratosphere (Smith et al., 2006; Steinwagner et al., 2010). Key constraints on the usefulness of isotope-enabled atmospheric and Earth system models include a lack of observed vertical profiles of the isotopic composition of water vapor and difficulties interpreting the influence of parameterized processes (e.g. Risi et al., 2012; Galewsky et al., 2016; Duan et al., 2018). Both constraints can be addressed to some extent by using isotope-enabled cloud resolving models (CRMs) and

LES models to simulate the detailed evolution of isotope composition in clouds and boundary layer processes over short time scales (Blossey et al., 2010; Eckstein et al., 2018; Wei et al., 2018; Risi et al., 2020). However, to date, few CRMs and LES models have enabled isotopic coupling together with mixed-phase microphysics, restricting the potential use of these models for investigating processes involved in cloud formation under the full range of atmospheric conditions. Additional tools for simulating boundary layer and cloud processes in the isotopic water cycle are needed to make best use of recent observational

isotopic datasets from large projects such as MOSAiC (Multidisciplinary drifting Observatory for the Study of Arctic Climate; Shupe et al., 2022; Nicolaus et al., 2022; Mellat et al., 2022) and MUSICA (MUlti-platform remote Sensing of Isotopologues for investigating the Cycle of Atmospheric water; Schneider et al., 2022). Such tools can help to bridge the gap between these observations and large-scale model simulations.

     In this paper, we introduce the isotope-enabled Python Cloud Large Eddy Simulation model (iPyCLES) with two-moment

mixed-phase microphysics. The base model, PyCLES (Pressel et al., 2015), was designed to address uncertainties in the behavior of stratocumulus clouds in global climate simulations and has been employed to study boundary-layer clouds in tropical and subtropical regions (Schneider et al., 2019; Shen et al., 2022) as well as the Arctic (Zhang et al., 2020) The iPyCLES extension implements isotopes as passive water tracers affected by dynamics, thermodynamics, boundary conditions, and microphysics in ways identical to water in the core model. To avoid limitations in the simulation of kinetic fractionation in mixed-phase

clouds associated with the use of saturation adjustment and a one-moment mixed-phase microphysical scheme in PyCLES, we further implement a comprehensive two-moment microphysical scheme based on Seifert and Beheng (2006a, b). In section 2, we describe the structure of the model and the isotopic formulation. In section 3, we present the results of three experimental cases conducted to assess the performance of iPyCLES, including non-precipitating tropical shallow convective cumulus, precipitating tropical shallow convective cumulus, and precipitating Arctic mixed-phase clouds. The results demonstrate both the

ability of iPyCLES to simulate the isotopic water cycle and the potential for isotopic signals to provide additional insights into cloud microphysics and mixing in tropical and Arctic boundary layers. In section 5, we outline some potential applications of iPyCLES for linking cloud microphysical processes with observable isotopic signals.

## 2   Model Description

### 2.1   Overview of the Model Structure

As introduced by Pressel et al. (2015), PyCLES is based on an energetically-consistent implementation of the anelastic equations of atmospheric motion that can be applied in simulations ranging from boundary layer dynamics (Batchelor, 1953; Dutton and Fichtl, 1969; Lipps and Hemler, 1982; Bannon, 1996) to deep convection (Pauluis, 2008). The total specific water content,



denoted as $q_t$, and specific entropy, denoted as $s$, are adopted as the primary prognostic variables. These prognostic variables are conserved during reversible adiabatic processes, providing a robust thermodynamic framework that is suitable for implementing isotopic tracers in a parallel passive water cycle without extensive changes to the dynamic and thermodynamic components of the model. Among multiple options for the model numerics, PyCLES includes several high-order weighted, essentially non-oscillatory (WENO) numerical schemes (Liu et al., 1994; Jiang and Shu, 1996). The WENO numerics reduce noise at coarser resolution (Pressel et al., 2015), thereby decreasing the computational cost of applying the model to convective development across the gray zone and allowing for the elimination of parameterized subgrid-scale (SGS) mixing above the surface layer (Pressel et al., 2017).

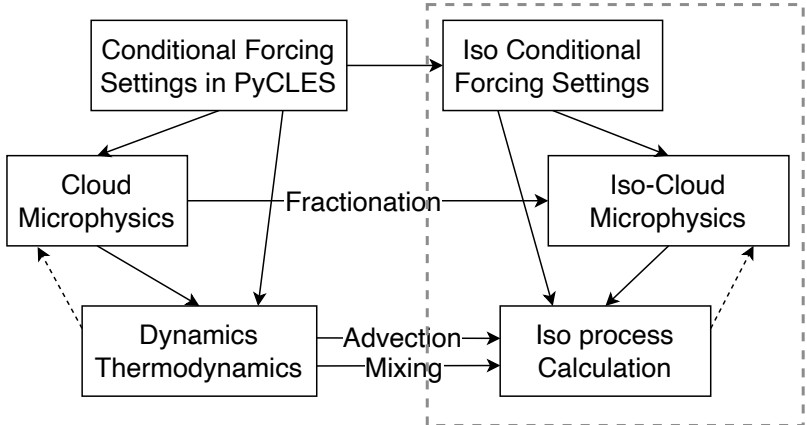

**Figure 1.** Diagram illustrating the implementation of passive isotopic water tracers (right column) in the LES model PyCLES (left column), rewrote based on Figure 1 in Wei et al. (2018). Solid arrows indicate concurrent processes while dashed arrows indicate processes computed sequentially.

Figure 1 illustrates the general design of the iPyCLES extension, highlighting the incorporation of passive isotopic water tracers (dashed box) and the links between the new isotopic module and the core dynamic and thermodynamic components of PyCLES (horizontal arrows). The three core components, including conditional settings and external forcing for moisture and heat, cloud microphysics, and dynamics and thermodynamics, are replicated in the isotopic water cycle while allowing for isotopic fractionation during phase changes.

Isotopic initialization and boundary conditions are flexible, with the default profile corresponding to Rayleigh distillation of an air parcel in equilibrium with conditions characteristic of the subtropical ocean. The dynamical component of the LES is responsible for computing isotopic tendencies due to scalar advection and SGS mixing. Isotopic concentrations in various water phases, including vapor, cloud droplets, cloud ice crystals, rain droplets, and snow, are simulated according to model-generated thermodynamic parameters, including the mass content of each water phase, ambient temperature, and reference atmospheric conditions such as pressure and air density. The most intricate aspects of the isotopic module are the computations

 

of fractionation during cloud microphysical processes, which are tightly interwoven with the model thermodynamics as detailed in section 2.4.

The mass balance of isotopologues within each water phase species encompasses mass advection, SGS turbulent mixing, and isotopic sources and sinks, as described in the mass balance equation:

$$\frac{\partial q^{\mathrm{iso}}}{\partial t} + \frac{1}{\rho_0}\frac{\partial\left(\rho_0 u_k q^{\mathrm{iso}}\right)}{\partial x_k} = -\frac{1}{\rho_0}\frac{\partial\left(\rho_0 \gamma_{q^{\mathrm{iso}},k}\right)}{\partial x_k} + \Phi_{\mathrm{iso}} \tag{1}$$

Here, $q^{\mathrm{iso}}$ denotes the isotope concentration in the specific humidity form, and $\gamma_{q^{\mathrm{iso}},k}$ is the $k$-th component of the SGS flux of the isotopic water species. The term $\Phi_{\mathrm{iso}}$ comprises the sources and sinks of each isotopic water species, including external forcing and surface fluxes along with cloud microphysical processes such as condensation, evaporation, and deposition:

$$\Phi_{\mathrm{iso}} = \left.\frac{\partial q^{\mathrm{iso}}}{\partial t}\right|_{\mathrm{SurfaceFlux}} + \left.\frac{\partial q^{\mathrm{iso}}}{\partial t}\right|_{\mathrm{ExternalForcing}} + \left.\frac{\partial q^{\mathrm{iso}}}{\partial t}\right|_{\mathrm{CloudMicrophysics}} \tag{2}$$

Further descriptions of tracer advection and SGS mixing in PyCLES have been provided by Pressel et al. (2015). The following sections outline the components of the model in greater detail.

## 2.2 Moist Thermodynamics

Temparature and other thermodynamic state variables are determined diagnostically from the prognostic variables total water and entropy, where the total water $q_t$ consists of vapor ($q_v$), liquid water ($q_l$), and ice ($q_i$) components:

$$q_t = q_v + q_l + q_i \tag{3}$$

and the specific entropy is the weighted sum of specific entropies corresponding to each component:

$$s = (1 - q_t)s_d + q_t s_v - q_l(s_v - s_l) - q_i(s_v - s_i), \tag{4}$$

with the subscript $d$ corresponding to dry air. These variables are then used to assess buoyancy and microphysical processes in cloud formation, as well as the sources and sinks of specific entropy (Pressel et al., 2015).

Phase partitioning under local thermodynamic equilibrium has a critical influence on the simulated isotopes. By default, PyCLES implements an empirical saturation adjustment method to diagnose the concentrations of cloud liquid water and ice (Grabowski, 1998; Kaul et al., 2015; Pressel et al., 2015). Under this approach, when air is unsaturated and in equilibrium, all water present is in the form of vapor ($q_t = q_v$). Three separate implementations are available in iPyCLES for treating conditions when the total water $q_t$ exceeds the saturation specific humidity $q_v^*(T,p)$. The first option, as described by Pressel et al. (2015), assumes that ice crystals form heterogeneously at temperatures below the freezing point $T_f$ and no supercooled liquid water is present (Grabowski, 1998; Kaul et al., 2015). The saturation specific humidity $q_v^*(T,p)$ is defined as the saturation specific humidity over liquid water $q_v^{*,l}(T,p)$ for temperatures larger than $T_f$ and as the saturation specific humidity $q_v^{*,i}(T,p)$





over ice for temperatures smaller than or equal to $T_f$:

$$q_v^*(T, p) = \begin{cases} q_v^{*,l} & \text{for} \quad T > T_f \\ q_v^{*,i} & \text{for} \quad T \leq T_f \end{cases} \tag{5}$$

When supersaturation occurs ($q_v > q_v^*(T, p)$), the amount of condensed water ($q_l$ or $q_i$) is equal to the saturation excess $\sigma$:

$$\sigma = (q_t - q_v^*)\,\mathcal{H}\,(q_t - q_v^*) \tag{6}$$

where the Heaviside step function $\mathcal{H}$ ensures supersaturation. The second option is a simplified one-moment mixed-phase microphysics scheme implemented by Zhang et al. (2020) following Kaul et al. (2015), as described in section 2.3.1. Under this option, supercooled liquid water and ice coexist at temperatures smaller the freezing point $T_f$ (Houghton, 1951; Korolev and Mazin, 2003; Field et al., 2004) and warmer than the homogeneous nucleation temperature $T_i$ (Straka, 2009, 233.15 K;). Phase partitioning is based on an empirical function $\lambda$ that depends on temperature:

$$\lambda(T) \begin{cases} 0 & T < T_i \\ \left(\frac{T - T_i}{T_f - T_i}\right)^n & T_i < T < T_f \\ 1 & T_f < T \end{cases} \tag{7}$$

and the specific liquid and ice water contents are diagnosed as:

$$q_l = \lambda(T)\sigma$$
$$q_i = [1 - \lambda(T)]\sigma \tag{8}$$

The empirical exponent $n$ is set to 0.1 for good agreement with observations and to ensure that $\lambda$ exceeds 80% across the entire range of mixed-phase conditions (Kaul et al., 2015). The third option, which eliminates saturation adjustment entirely, implements the two-moment mixed-phase microphysics scheme proposed by (Seifert and Beheng, 2006b) and is described in section 2.3.2. Further details on the calculation of saturation specific humidities and other aspects of the moist thermodynamics have been provided by Pressel et al. (2015).

## 2.3 Cloud Microphysics

Two different bulk microphysics schemes have previously been implemented in PyCLES. The first, which we refer to as SBWarm, is a warm-phase-only version of the two-moment microphysical scheme of Seifert and Beheng (2006b) with modifications as outlined by Tan et al. (2016). The second, referred to as Kaul15, is a one-moment mixed-phase scheme based on Kaul et al. (2015) that was implemented in PyCLES by Zhang et al. (2020). Both schemes use the mixed-phase saturation adjustment procedure desribed in equations (7)–(8) to diagnostically partition cloud liquid water and ice. To facilitate a fuller exploration of isotopic exchange in mixed-phase clouds, particularly those processes that involve significant kinetic fractionation during deposition to ice (Jouzel and Merlivat, 1984; Galewsky et al., 2011; Samuels-Crow et al., 2014), we extend the





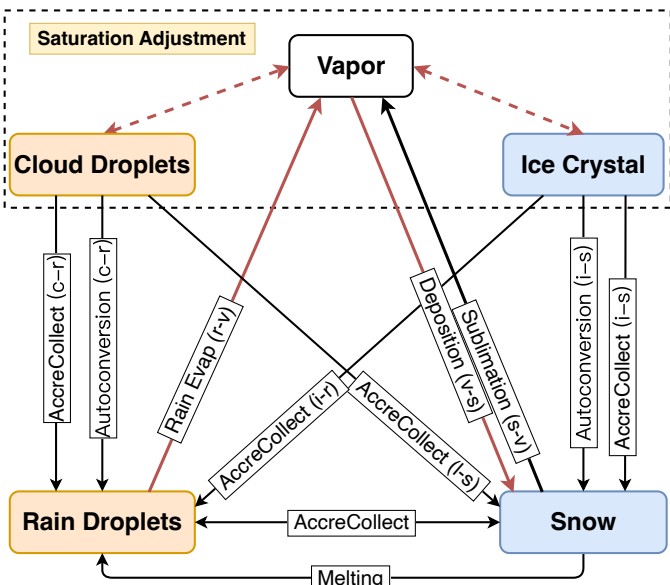

**Figure 2.** Schematic diagram of the one-moment bulk microphysics scheme Kaul15. Red solid lines with arrows indicate microphysical processes that may involve isotopic fractionation during phase changes. Red dashed lines represent isotopically relevant processes treated under mixed-phase saturation adjustment.

SBWarm scheme to include the parameterizations of ice and snow microphysics proposed by Seifert and Beheng (2006b). This extension yields a two-moment mixed-phase cloud microphysics without saturation adjustment, which we refer to as SB06. As SBWarm is included within SB06, only the mixed-phase version is described below. Isotopic tracers are implemented in all three microphysical schemes as described in section 2.4.2.

### 2.3.1 One-Moment Mixed-Phase Microphysics

In the Kaul15 scheme, cloud specific liquid water content $q_l$ and cloud specific ice water content $q_i$ are diagnosed based on saturation adjustment (section 2.2). By contrast, specific rain water content $q_r$ and specific snow water content $q_s$ are prognostic variables calculated by the cloud microphysics module.

Figure 2 illustrates how the Kaul15 scheme represents exchanges among microphysical species during cloud processes. Prognostic exchange between vapor and rain or snow is parameterized in Kaul15 as:

$$\frac{\partial q_k}{\partial t}_{\text{evp,dep}} = AS_k[a_k + b_k \text{Re}_k^{1/2}]G_k(T)N_{0,k}\lambda_k^{-2} \tag{9}$$

where $k$ refers to rain or snow, $\text{Re}_k$ is the corresponding Reynolds number, and $S_k = q/q^* - 1$ is the corresponding super-saturation ratio. Positive values of $S_k$ indicate condensation or deposition while negative values indicate evaporation or sublimation. The shape parameter $A_r = 2\pi$ for rain and $A_s = \frac{4}{3}\pi$ for snow, while the ventilation parameters are set to $a_r = 0.78$ and $b_r = 0.27$ for rain and to $a_s = 0.65$ and $b_s = 0.39$ for snow. Following Straka (2009), Zhang et al. (2020) revised the





temperature-dependent thermodynamic function $G_k(T)$, which controls the hydrometeor growth rate:

$$G_k(T) = \left[ \frac{\rho_k R_v T}{D p_v^*(T)} + \left( \frac{L_k}{R_v T} - 1 \right) \frac{L \rho_k}{\kappa T} \right]^{-1} \tag{10}$$

where $\rho_k$ is the density of the corresponding species, $R_v$ is the gas constant for water vapor, $L_k$ is the latent heat of vaporization ($L_v$) or sublimation ($L_s$), $p_v^*(T,p)$ is the saturation vapor pressure over liquid water or ice, $\kappa = 2.5 \times 10^{-2}\,\mathrm{J\,m^{-1}\,s^{-1}\,K^{-1}}$ is the thermal conductivity of air, and $D$ is the diffusivity of water vapor. A melting parameterization is included to describe the transition of snow to raindrops, leading to a sink of $q_s$ and a source of $q_r$. The mass balance framework in PyCLES removes prognostic precipitation species after they are generated, leading to a loss term in the total specific water content:

$$\frac{\partial q_t}{\partial t}_{\mathrm{micro}} = - \left( \frac{\partial q_r}{\partial t} + \frac{\partial q_s}{\partial t} \right) \tag{11}$$

Further details of this parameterization and its performance have been provided by Kaul et al. (2015), Zhang et al. (2020), and references therein.

### 2.3.2 Two-Moment Mixed-Phase Microphysics

The SB06 two-moment microphysics scheme was initially developed to improve the portrayal of mixed-phase clouds in a mesoscale atmospheric model (Seifert and Beheng, 2006b). This parameterization has several notable benefits for CRMs and LES models, such as the ability to handle short timesteps under non-equilibrium conditions and the ability to account for turbulent effects on droplet coalescence (Stevens et al., 2005; Vogel et al., 2009). The SB06 representation of liquid-only processes was adopted early in PyCLES to represent interactions among water vapor, cloud droplets, and rain droplets. In this work, we have extended the SB06 scheme in PyCLES to represent the formation and evolution of ice and snow as well.

Figure 3 illustrates the generation of and transformations among four hydrometer species and water vapor as represented in the SB06 scheme. Rain evaporation follows Pruppacher and Klett (1997) with the modification suggested by Seifert (2008):

$$\frac{\partial q_r}{\partial t}_{\mathrm{evp}} = 2\pi n_r G_r(T) S_l \overline{F}_v$$
$$\frac{\partial n_r}{\partial t}_{\mathrm{evp}} = \gamma_r m_r \frac{\partial q_r}{\partial t}_{\mathrm{evp}}. \tag{12}$$

Here $S_l$ is the supersaturation ratio over liquid water, $\overline{F}_v$ is the average ventilation factor (Seifert, 2008, Eqs. A6-A7), $n_r$ is the number of raindrops, $m_r$ is the mass of raindrops, and the coefficient $\gamma_r$ is set to be 0.7 following Seifert (2008). This setting matches that used in the Dutch Atmospheric Large-Eddy Simulation (DALES) model (Heus et al., 2010). The thermodynamic factor $G_r(T)$ for rain is defined as:

$$G_r(T) = \left[ \frac{R_v T}{p_v^{*,l} D_v} + \frac{L_v}{\kappa T} \left( \frac{L_v}{R_v T} - 1 \right) \right]^{-1} \tag{13}$$

with $L_v$ the latent heat of vaporization. Deposition and sublimation of cloud ice and snow are represented following Cotton et al. (1986) and Pruppacher and Klett (1997):

$$\frac{\partial q_{i,s}}{\partial t}_{\mathrm{dep,sub}} = \frac{4\pi}{c_{i,s}} G_i(T) D_i \overline{F}_{v,i} S_i. \tag{14}$$



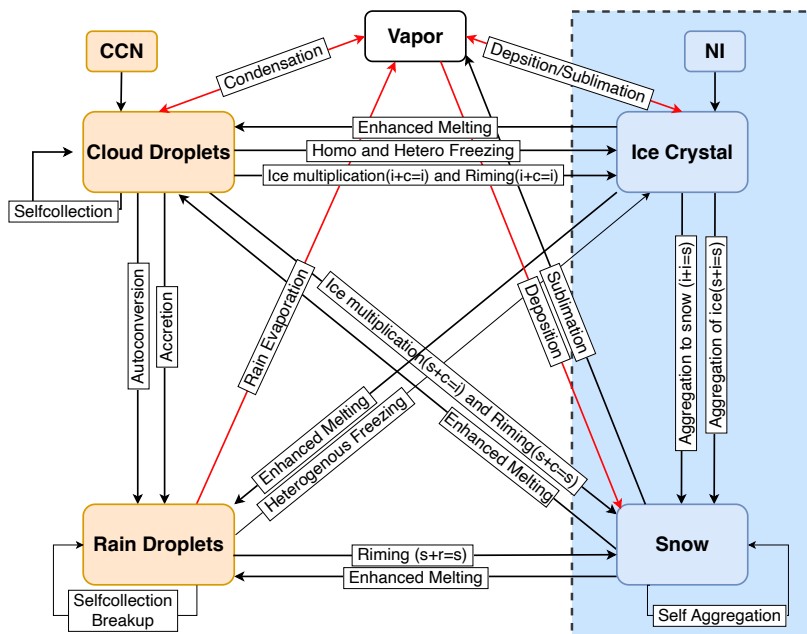

**Figure 3.** Schematic diagram of the mixed-phase SB06 scheme with species including cloud droplets, cloud ice, rain, and snow. Red arrows indicate exchange processes that involve isotopic fractionation during phase changes. Mixed-phase saturation adjustment is replaced by activation of cloud droplets and nucleation, deposition, and sublimation of ice crystals.

Here, $c_i = \pi$ is for spherical cloud ice particles and $c_s = 2$ is for snow, $\overline{F}_{v,i}$ is the average ventilation factor for ice deposition (Seifert and Beheng, 2006b, their Appendix B), and $S_i$ is the supersaturation ratio over ice, indicating whether the process results in a source ($S_i < 0$) or sink ($S_i > 0$) of water vapor. The thermodynamic factor $G_i(T)$ follows the same expression as $G_r(T)$ (eq. (13)) but with $p_v^{*,l}$ replaced by $p_v^{*,i}$ and $L_v$ replaced by $L_s$, the latent heat of sublimation.

## 2.4 Isotope Module

### 2.4.1 Isotopic Tracers in iPyCLES

Two stable water isotopologues, $H_2^{18}O$ and HDO (i.e. $^1H^2H^{16}O$), are implemented in the current version of iPyCLES. Both sets of isotopic tracers are implemented as prognostic variables following the budget equation for isotopic species (eq. (1)). Isotopic species include the isotopic specific contents of water vapor ($q_v^{\text{iso}}$), cloud liquid water ($q_l^{\text{iso}}$), cloud ice ($q_i^{\text{iso}}$), rain ($q_r^{\text{iso}}$), and snow ($q_s^{\text{iso}}$). An isotopic total water tracer $q_t^{\text{iso}}$ is also tracked and compared to the sum $q_v^{\text{iso}} + q_l^{\text{iso}} + q_i^{\text{iso}}$ to ensure internal consistency. Species corresponding to each isotope (i.e. $H_2^{18}O$ and HDO) evolve independently. The definition of $q^{\text{iso}}$ is analogous to that of $q$, namely the mass content of the isotope in the specified water species relative to the mass of moist air.

It is common practice to report isotopic ratios as $\delta$ values representing depletion relative to a fixed standard:

$$\delta = \left( \frac{R}{R_{\text{std}}} - 1 \right) \times 1000\%o \tag{15}$$





where $R$ is the mass ratio of the rare isotope relative to common water ($^2\text{H}_2^{16}\text{O}$) and $R_{\text{std}}$ is the corresponding ratio in an accepted standard. In iPyCLES we take the standard ratio $R_{\text{std}}$ for each isotope to be that in Vienna Standard Mean Ocean Water (V-SMOW) and calculate $R$ as the ratio of $q^{\text{iso}}$ and $q$:

$$R_{\text{iso}} = \frac{q_{\text{iso}}}{q}. \tag{16}$$

The use of $\delta$ values is motivated by the smallness of fluctuations in $R$.

The source and sink term for isotopic concentrations in each water species, $\Phi_{\text{iso}}$, may be influenced by coupling with cloud microphysics, surface fluxes, and any specified external forcing. Tendencies due to surface fluxes and external forcing are applied only to $q_t^{\text{iso}}$ and $q_v^{\text{iso}}$. Isotopes in rain and snow sediment according to the fall velocities computed for rain and snow in

the microphysics scheme.

Isotopic exchange via the saturation adjustment method for condensation and evaporation of cloud liquid water at thermodynamic equilibrium is represented as equilibrium fractionation following the diagnostic approach outlined by Ciais and Jouzel (1994). In this approach, which applies to all microphysical schemes implemented in iPyCLES, $q_v^{\text{iso}}$ and $q_l^{\text{iso}}$ are calculated by adjusting the partitioning of vapor and cloud liquid water from the LES computation to account for temperature-dependent

equilibrium fractionation:

$$q_l^{iso} = \frac{q_{lv}^{\text{iso}}}{1 + \frac{q_v}{q_l} \times \frac{1}{\alpha_{lv}}} \tag{17}$$

Here, $q_{lv}^{\text{iso}}$ is the sum of the isotopic specific contents of liquid water and water vapor (i.e. $q_{lv}^{\text{iso}} = q_t^{\text{iso}} - q_i^{\text{iso}}$) and $\alpha_{lv}$ is the temperature-dependent equilibrium fractionation factor between liquid water and water vapor (Majoube, 1971):

$$\alpha_{lv} = \frac{R_{\text{liq}}}{R_{\text{vap}}} = \exp\left(aT^{-2} + bT^{-1} + c\right) \begin{cases} a = 24844, b = \text{-}76.248, c = 52.61 \times 10^{-3} & \text{HDO/H}_2\text{O} \\ a = 1137, b = \text{-}0.4156, c = -2.0667 \times 10^{-3} & \text{H}_2^{18}\text{O/H}_2\text{O} \end{cases} \tag{18}$$

The specific humidity of the water vapor isotope, $q_v^{\text{iso}}$, is then computed as $q_v^{\text{iso}} = q_{lv}^{\text{iso}} - q_l^{\text{iso}}$.

Isotopic exchange also occurs during phase transitions when thermodynamic equilibrium does not apply. Non-equilibrium or kinetic fractionation may occur during deposition to ice in mixed- and solid-phase clouds or evaporation of raindrops in unsaturated environments. During kinetic fractionation, the isotopic fluxes of $\text{H}_2^{18}\text{O}$ and HDO diverge from each other due to the different molecular masses and diffusivities of each isotope (Jouzel and Merlivat, 1984; Rozanski et al., 2001). The

280 cumulative extent of this mismatch can be measured by the deuterium excess d:

$$\text{d} = \delta\text{D} - 8 \cdot \delta^{18}\text{O}(‰), \tag{19}$$

where $\delta\text{D}$ denotes the isotopic $\delta$ ratio of HDO, $\delta^{18}O$ denotes that of $\text{H}_2^{18}\text{O}$, and the factor 8 derives from the global meteoric water line (Dansgaard, 1964; Gat et al., 1996). A higher deuterium excess d indicates a greater concentration of HDO relative to $\text{H}_2^{18}\text{O}$, and vice versa. Kinetic fractionation factors associated with cloud and surface processes are introduced in the following

two sections.



### 2.4.2 Coupling with Cloud Microphysics

The isotope module is coupled with the cloud microphysics scheme to compute exchanges of isotopic mass among different water species. The isotopic module is fully integrated into both the SB06 two-moment microphysics scheme (including SBWarm) and the Kaul15 one-moment microphysics scheme, and applies to processes both with and without isotopic fractionation.

The isotopic specific water content tendencies due to microphysical processes for cloud liquid water, cloud ice, rain, and snow can be expressed as:

$$
\frac{\partial q_l^{\mathrm{iso}}}{\partial t} = \left.\frac{\partial q_l^{\mathrm{iso}}}{\partial t}\right|_{\mathrm{act}} - \left.\frac{\partial q_l^{\mathrm{iso}}}{\partial t}\right|_{\mathrm{lve}} - \left.\frac{\partial q_i^{\mathrm{iso}}}{\partial t}\right|_{\mathrm{frz}} - \left.\frac{\partial q_i^{\mathrm{iso}}}{\partial t}\right|_{\mathrm{rime}} - \left.\frac{\partial q_{ls}^{\mathrm{iso}}}{\partial t}\right|_{\mathrm{rime}} - \left.\frac{\partial q_r^{\mathrm{iso}}}{\partial t}\right|_{\mathrm{aut}} - \left.\frac{\partial q_r^{\mathrm{iso}}}{\partial t}\right|_{\mathrm{acc}}
$$

$$
\frac{\partial q_i^{\mathrm{iso}}}{\partial t} = \left.\frac{\partial q_i^{\mathrm{iso}}}{\partial t}\right|_{\mathrm{nuc}} + \left.\frac{\partial q_i^{\mathrm{iso}}}{\partial t}\right|_{\mathrm{dep}} - \left.\frac{\partial q_i^{\mathrm{iso}}}{\partial t}\right|_{\mathrm{sub}} + \left.\frac{\partial q_i^{\mathrm{iso}}}{\partial t}\right|_{\mathrm{frz}} + \left.\frac{\partial q_i^{\mathrm{iso}}}{\partial t}\right|_{\mathrm{rime}} - \left.\frac{\partial q_s^{\mathrm{iso}}}{\partial t}\right|_{\mathrm{agg}}
$$

$$
\frac{\partial q_r^{\mathrm{iso}}}{\partial t} = \left.\frac{\partial q_r^{\mathrm{iso}}}{\partial t}\right|_{\mathrm{aut}} + \left.\frac{\partial q_r^{\mathrm{iso}}}{\partial t}\right|_{\mathrm{acc}} - \left.\frac{\partial q_r^{\mathrm{iso}}}{\partial t}\right|_{\mathrm{lve}} - \left.\frac{\partial q_r^{\mathrm{iso}}}{\partial t}\right|_{\mathrm{kinevp}} - \left.\frac{\partial q_s^{\mathrm{iso}}}{\partial t}\right|_{\mathrm{frz}} - \left.\frac{\partial q_{rs}^{\mathrm{iso}}}{\partial t}\right|_{\mathrm{rime}} + \left.\frac{\partial q_s^{\mathrm{iso}}}{\partial t}\right|_{\mathrm{mlt}}
$$

$$
\frac{\partial q_s^{\mathrm{iso}}}{\partial t} = \left.\frac{\partial q_s^{\mathrm{iso}}}{\partial t}\right|_{\mathrm{agg}} + \left.\frac{\partial q_s^{\mathrm{iso}}}{\partial t}\right|_{\mathrm{dep}} - \left.\frac{\partial q_s^{\mathrm{iso}}}{\partial t}\right|_{\mathrm{sub}} + \left.\frac{\partial q_{ls}^{\mathrm{iso}}}{\partial t}\right|_{\mathrm{rime}} + \left.\frac{\partial q_{rs}^{\mathrm{iso}}}{\partial t}\right|_{\mathrm{rime}} - \left.\frac{\partial q_s^{\mathrm{iso}}}{\partial t}\right|_{\mathrm{mlt}}
\tag{20}
$$

Where the subscript 'act' refers to activation, 'lve' to liquid-vapor exchange at thermodynamic equilibrium (which may or may not be associated with net condensation or evaporation), 'frz' to freezing, 'rime' to riming, 'aut' to autoconversion, 'acc' to accretion, 'nuc' to nucleation, 'dep' to deposition of vapor to ice, 'sub' to sublimation of ice to vapor, 'agg' to aggregation, 'kinevp' to kinetic evaporation of raindrops in unsaturated environments, and 'mlt' to melting. The tendency equation for isotopes in water vapor in terms of the condensed species is then:

$$
\frac{\partial q_v^{\mathrm{iso}}}{\partial t} = \underbrace{\left.\frac{\partial q_l^{\mathrm{iso}}}{\partial t}\right|_{\mathrm{lve}} - \left.\frac{\partial q_l^{\mathrm{iso}}}{\partial t}\right|_{\mathrm{act}} + \left.\frac{\partial q_r^{\mathrm{iso}}}{\partial t}\right|_{\mathrm{lve}}}_{\mathrm{equilibrium}} - \underbrace{\left.\frac{\partial q_i^{\mathrm{iso}}}{\partial t}\right|_{\mathrm{nuc}} - \left.\frac{\partial q_i^{\mathrm{iso}}}{\partial t}\right|_{\mathrm{dep}} + \left.\frac{\partial q_r^{\mathrm{iso}}}{\partial t}\right|_{\mathrm{kinevp}} - \left.\frac{\partial q_s^{\mathrm{iso}}}{\partial t}\right|_{\mathrm{dep}}}_{\mathrm{kinetic}} + \underbrace{\left.\frac{\partial q_i^{\mathrm{iso}}}{\partial t}\right|_{\mathrm{sub}} + \left.\frac{\partial q_s^{\mathrm{iso}}}{\partial t}\right|_{\mathrm{sub}}}_{\mathrm{non-fractionating}}
\tag{21}
$$

where processes involving equilibrium fractionation, kinetic fractionation, and no fractionation are labeled. Isotopic fractionation is computed for all processed depicted as red arrows in Figures 2 and 3, which all involve vapor–liquid or vapor–ice exchange. Sublimation of ice particles is assumed to occur without fractionation because of the low diffusivity of water molecules in ice (Bony et al., 2008). The total isotopic specific humidity $q_t^{\mathrm{iso}}$ is modified by the irreversible precipitation source:

$$
\frac{\partial q_t^{\mathrm{iso}}}{\partial t} = -\left( \frac{\partial q_r^{\mathrm{iso}}}{\partial t} + \frac{\partial q_s^{\mathrm{iso}}}{\partial t} \right)
\tag{22}
$$

Other processes, such as autoconversion, accretion, riming, aggregation, and melting, represent conversions among condensed phase species and are treated as non-fractionating processes by the isotope module. The isotopic tendency for a non-fractionating processes, such as autoconversion, is calculated based on the corresponding specific humidity tendency from the microphysics scheme and the isotope ratio of the source species $R_{\mathrm{src}}$:

$$
\left.\frac{\partial q^{\mathrm{iso}}}{\partial t}\right|_{\mathrm{aut}} = \left.\frac{\partial q}{\partial t}\right|_{\mathrm{aut}} \times R_{\mathrm{src}}
\tag{23}
$$



Similar equations apply for all exchanges depicted as black arrows in Figures 2 and 3, as well as the sublimation of ice.

Isotopic exchange between liquid water and water vapor is typically considered to involve equilibrium fractionation owing to the rapid relaxation time between vapor and cloud liquid droplets (Jouzel and Merlivat, 1984; Ciais and Jouzel, 1994; Bolot et al., 2013). Consistent with the continued application of saturation adjustment for evaporation and condensation of cloud droplets in both microphysical schemes, activation, condensation, and evaporation of cloud drops are calculated using the equilibrium fractionation factor (eq. (17)). Water vapor is also assumed to equilibrate with cloud droplets even when there is

no net condensation or evaporation, and to partially equilibrate with raindrops, with the extent of equilibration dependent on the raindrop fall speed (e.g. Lee and Fung, 2008).

Kinetic fractionation associated with the evaporation of falling raindrops is based on the parameterization proposed by Blossey et al. (2010, their eqs. B17-B19):

$$\frac{\partial q_r^{\mathrm{iso}}}{\partial t}_{\mathrm{kinevp}} = 2\pi D_r \overline{F}_v^{\mathrm{iso}} n_r G(T)^{\mathrm{iso}} \tag{24}$$

where $D_r$ is the average diameter of raindrops in the bulk microphysical scheme, $n_r$ is the number density of raindrops, and $\overline{F}_v^{\mathrm{iso}}$ is the ventilation factor for the corresponding isotope. The thermodynamic function $G(T)^{\mathrm{iso}}$ is calculated from thermodynamic conditions as proposed by Blossey et al. (2010):

$$G(T)^{\mathrm{iso}} = D_v^{\mathrm{iso}} \left[ \frac{q_r^{\mathrm{iso}}}{q_r} \frac{1}{\alpha_{lv}} \times \left( \frac{1 + b_l(S_l + 1)}{1 + b_l} \right) - \frac{q_v^{\mathrm{iso}}}{q_v} \times (S_l + 1) \right] \rho_v^{*,l}\left[ T(\infty), p \right], \tag{25}$$

Here, $D_v^{\mathrm{iso}}$ is the diffusivity of the water vapor isotope and $\rho_v^{*,l} = p_v^{*,l}/R_v T$ is computed from the ideal gas law for water

vapor at the ambient temperature $T(\infty)$. The coefficient $b_l$ is modified in iPyCLES to match the parameterization of $G_r(T)$ (eq. (13)):

$$b_l \equiv D_v \rho_v^{*,l} \frac{L_v}{\kappa T} \left( \frac{L_v}{R_v T} - 1 \right). \tag{26}$$

Ratios of the diffusivity of rare isotopes to the diffusivity of the most abundant isotope are available from several sources in iPyCLES. For the simulations below, these ratios are adopted from Cappa et al. (2003) as $D^{\mathrm{H}_2^{18}\mathrm{O}}/D^{\mathrm{H}_2\mathrm{O}} = 0.9691$ and

$D^{\mathrm{HDO}}/D^{\mathrm{H}_2\mathrm{O}} = 0.9839$. Equation (25) for $G(T)^{\mathrm{iso}}$ is also used in the Kaul15 scheme in conjunction with eq. (9) to calculate kinetic fractionation during rain evaporation.

Kinetic fractionation also occurs in vapor diffusion over ice phase particles such as cloud ice crystals and snow, with important effects on deuterium excess (Jouzel and Merlivat, 1984; Blossey et al., 2010). This parameterization scheme also follows that proposed by Blossey et al. (2010, their eqs. B25-B26):

$$\frac{\partial q_i^{\mathrm{iso}}}{\partial t} = \alpha_{iv} \alpha_{k,i} \frac{q_v^{\mathrm{iso}}}{q_v} \frac{\partial q_i}{\partial t}, \tag{27}$$

where $\frac{\partial q_i}{\partial t}$ is the deposition tendency of vapor specific humidity on ice particles, $\alpha_{iv}$ is the equilibrium fractionation factor between ice and vapor, and $\alpha_{k,i}$ is the kinetic fractionation factor during deposition. In iPyCLES, the equilibrium fractionation factor $\alpha_{iv}$ is calculated using the empirical expression suggested by Ellehoj et al. (2013):

$$\alpha_{iv} = \frac{R_{\mathrm{ice}}}{R_{\mathrm{vap}}} = \exp\left( aT^{-2} + bT^{-1} + c \right) \begin{cases} a = 48888, b = \text{-}203.10, c = 0.2133 & \mathrm{HDO/H_2O} \\ a = 8312.5, b = \text{-}49.192, c = 0.0831 & \mathrm{H_2^{18}O/H_2O} \end{cases}, \tag{28}$$





while the kinetic fractionation factor $\alpha_{k,i}$ calculated as (Blossey et al., 2010):

$$\alpha_{k,i} = \frac{(1+b_i)(S_i+1)}{\alpha_{iv}\frac{\overline{F}_v D_v}{\overline{F}_v^{\mathrm{iso}} D_v^{\mathrm{iso}}}S_i + [1 + b_i(S_i+1)]}, \tag{29}$$

where $b_i$ is

$$b_i \equiv \frac{D_v (L_v + L_s)^2 \rho_v^{*,i}}{\kappa R_v T^2}. \tag{30}$$

Here, $S_i$ is the supersaturation ratio over ice, $\overline{F}_v$ and $\overline{F}_v^{\mathrm{iso}}$ are the ventilation factors for the abundant and rare isotopes, respectively, and $\rho_v^{*,i}$ is the saturation vapor density over ice relative to the ambient air temperature. This expression for kinetic isotope exchange between water vapor and ice also applies to ice nucleation in the mixed-phase cloud regime, including exchange associated with the Wegener–Bergeron–Findeisen process (e.g. Korolev and Mazin, 2003).

The tendency equations for the evolution of isotopes in precipitation are simplified in the Kaul15 scheme relative to those in SB06:

$$\frac{\partial q_r^{\mathrm{iso}}}{\partial t} = \left.\frac{\partial q_r^{\mathrm{iso}}}{\partial t}\right|_{\mathrm{aut}} + \left.\frac{\partial q_r^{\mathrm{iso}}}{\partial t}\right|_{\mathrm{acc}} + \left.\frac{\partial q_s^{\mathrm{iso}}}{\partial t}\right|_{\mathrm{mlt}} - \left.\frac{\partial q_r^{\mathrm{iso}}}{\partial t}\right|_{\mathrm{evp}}$$

$$\frac{\partial q_s^{\mathrm{iso}}}{\partial t} = \left.\frac{\partial q_s^{\mathrm{iso}}}{\partial t}\right|_{\mathrm{aut}} + \left.\frac{\partial q_s^{\mathrm{iso}}}{\partial t}\right|_{\mathrm{acc}} + \left.\frac{\partial q_s^{\mathrm{iso}}}{\partial t}\right|_{\mathrm{dep}} - \left.\frac{\partial q_s^{\mathrm{iso}}}{\partial t}\right|_{\mathrm{mlt}} - \left.\frac{\partial q_s^{\mathrm{iso}}}{\partial t}\right|_{\mathrm{sub}} \tag{31}$$

Isotopic exchange associated with rain evaporation, liquid-vapor exchange, and non-fractionating processes are treated identically in the Kaul15 scheme as in the SB06 scheme. The mixed-phase saturation adjustment procedure in PyCLES coupled with the one-moment Kaul15 microphysics scheme diagnoses cloud ice concentration as outlined in eq. (8). To implement the isotopic fractionation parameterization for vapor deposition to ice (eq. (27)) in Kaul15, the change in cloud ice concentration is computed across each timestep:

$$\frac{\partial q_i}{\partial t} \approx \frac{\Delta q_i}{\Delta t} = \frac{q_i(t) - q_i(t + \Delta t)}{\Delta t}, \tag{32}$$

where $\Delta t$ is the model timestep.

In addition to the mixed-phase implementation of SB06, all previously implemented microphysics schemes (including the warm-phase microphysics based on SB06 and the no-microphysics option) are retained in iPyCLES and can be specified to match different simulation objectives.

### 2.4.3 Surface Sources and External Forcing

The source and sink term in the tendency equation for isotopic tracers (eq. (2)) also includes the effects of surface evaporation and specified external forcings on the isotope budget. Isotopic tendencies due to specified external forcing are non-fractionating. For example, tendencies due to prescribed subsidence are computed based on the specific humidity and isotopic ratio of the overlying moisture:

$$\left.\frac{\partial q^{\mathrm{iso}}}{\partial t}\right|_{\mathrm{Subsidence}} = \frac{(R_f q_f - Rq)}{\Delta z} w_{\mathrm{Subsidence}} \tag{33}$$





For subsidence, $R$ and $q$ are the local isotopic ratio and specific humidity and $R_f$ and $q_f$ are the isotopic ratio and specific humidity in the overlying grid cell, i.e.:

$$R_f = \frac{q_{z+1}^{\mathrm{iso}}}{q_{z+1}} \tag{34}$$

The isotopic effects of large-scale advective forcing from outside the domain are implemented in a similar way, with $R_f$ and $q_f$ at each model level specified in the simulation settings.

Surface fluxes set the lower boundary condition for the isotopic tracers. In the standard configuration of PyCLES, the surface flux of total water is assumed to originate as evaporation from an exposed water surface. The isotopic surface flux can be determined by computing the evaporation flux of total water and the isotopic ratio in the evaporation flux $R_{\mathrm{fx}}$:

$$\frac{\partial q_t^{\mathrm{iso}}}{\partial t}\bigg|_{\mathrm{SurfaceFlux}} = R_{\mathrm{fx}} \frac{\partial q_t}{\partial t}\bigg|_{\mathrm{SurfaceFlux}} \tag{35}$$

The ratio $R_{\mathrm{fx}}$ is calculated based on a revised version of the Craig–Gordon model (Craig and Gordon, 1965) proposed by Merlivat (1978):

$$R_{\mathrm{fx}} = \frac{\alpha_{lv}\alpha_k R_{\mathrm{src}}}{(1-\mathrm{RH}) + \alpha_k \cdot \mathrm{RH}}, \tag{36}$$

where RH is the surface layer relative humidity, $R_{\mathrm{src}}$ is the isotopic ratio in the source water, and $\alpha_k$ is a kinetic fractionation

factor that depends on the relative humidity and surface roughness:

$$\alpha_k = 1 - n\left(1 - \mathrm{RH}\right)\left(1 - \frac{D_v^{\mathrm{iso}}}{D_v}\right) \tag{37}$$

The value of the coefficient $n$ is set to $n = 2/3$ for smooth surfaces and $n = 1/2$ for rough surfaces (Horita et al., 2008). Kinetic fractionation is therefore most pronounced for dry surface air overlying a smooth surface. Additional details regarding this formulation of the Craig–Gordon model have been provided by Horita et al. (2008).

## 3   Model Evaluation

We evaluate the performance of iPyCLES in simulating cloud fields and isotopes under varying conditionsis by conducting three test cases for stratocumulus clouds in subtropical and Arctic environments. The first subtropical test case focuses on a non-precipitatinng stratocumulus case from the BOMEX (Barbados Oceanographic and Meteorological EXperiment; Kuettner and Holland, 1969) campaign using saturation adjustment without microphysics. The second subtropical test case focuses on

a precipitating stratocumulus case from the RICO (Rain in Cumulus Over the Ocean; Rauber et al., 2007) campaign using the SBWarm microphysics with saturation adjustment and $\lambda = 1.0$. The Arctic test case focuses on a mixed-phase stratocumulus cloud observed during the ISDAC (Indirect and Semi-Direct Aerosol Campaign McFarquhar et al., 2011). Separate simulations are conducted for the Arctic test case using both the Kaul15 and SB06 microphysics schemes to facilitate evaluation and intercomparison of simulated cloud fields and isotopic ratios between the one-moment and two-moment mixed-phase schemes.

Test case names and key settings, including grid spacing, grid size, and parameterization choices, are listed in Table 1. All



**Table 1.** List of test cases carried out using iPyCLES, including grid spacing, grid size, microphysical parameterization, and surface isotopic source. The C-G model is coupled based on eq. (36).

| Name | $\Delta x = \Delta y$ [m] | $\Delta z$ [m] | Grid ($x \times y \times z$) | Microphysics | Surface |
|------|------|------|------|------|------|
| BOMEX | 100 | 100 | 64×64×75 | None | C-G |
| RICO | 100 | 40 | 128×128×125 | SBWarm | C-G |
| ISDAC-1M | 50 | 10 | 64×64×250 | Kaul15 | None |
| ISDAC-2M | 50 | 10 | 64×64×250 | SB06 | None |

numerical solutions are obtained using the fifth-order WENO numerics in PyCLES (Pressel et al., 2015, 2017; Tan et al., 2018).

Initial profiles of isotopic concentration are set using an idealized Rayleigh distillation formula to determine the vertical distribution of $\delta^{18}$O (Lee et al., 2006, 2012b):

$$\delta^{18}\text{O} = 8.99 \ln\left(q_v/0.622\right) - 42.9, \tag{38}$$

where $q_v$ is the water vapor specific humidity in units of $\text{g kg}^{-1}$. The profile of $\delta$D is initialized based on the global meteoric water line assuming a deuterium excess d of 10‰ (Craig, 1961; Merlivat and Jouzel, 1979):

$$\delta\text{D} = 8 \times \delta^{18}\text{O} + 10. \tag{39}$$

Although this assumption of constant d in water vapor at the initial time is inconsistent with observations (cf. Wei et al., 2018, their Fig. 3), it allows us to readily distinguish the effects of cloud processes in the results relative to vertical mixing of the initial profile.

### 3.1 The BOMEX Case: Non-precipitating Subtropical Cumulus

To assess the performance of the isotopic module with respect to the model dynamics and thermodynamics without microphysics, we have integrated the isotope component within a common test case for subtropical non-precipitating cumulus. The BOMEX case represents a prototypical cumulus-topped marine boundary layer. It has been widely used in LES simulations for investigating tropical cloud formation and boundary layer mass flux (Tiedtke, 1989; Salzen and McFarlane, 2002; Jiang and Cotton, 2000; de Roode and Bretherton, 2003; Nie and Kuang, 2012) and validating representations of shallow convective in larger-scale models (Tan et al., 2018; McIntyre et al., 2022; Janssens et al., 2023). Tan et al. (2018) previously simulated the BOMEX case in PyCLES, using the LES result to assess the performance of a single-column extended eddy-diffusivity mass-flux boundary layer scheme. The case settings derive from an LES intercomparison study conducted by Siebesma et al. (2003).

Figure 4 shows the initial profiles of dynamic and thermodynamic components for the BOMEX case. No liquid clouds are present at the beginning of the integration. The initial isotopic concentration of the total water is based on Rayleigh distillation and a constant deuterium excess of 10‰, as shown in Figure 4c. The initial state is well-mixed below 500 m, with an unstable




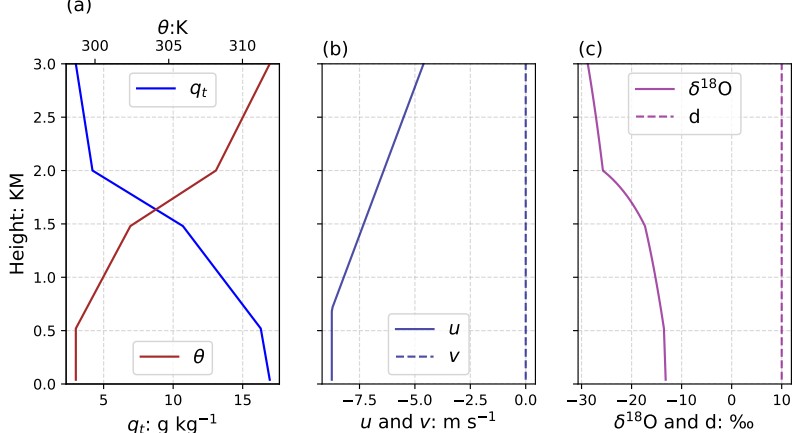

**Figure 4.** Initial vertical profiles of (a) total specific water content $q_t$ (where $q_v = q_t$ at the initial time) and potential temperature $\theta$; (b) the horizontal wind components $u$ and $v$; and (c) the isotopic ratio $\delta^{18}$O and the deuterium excess d for the BOMEX simulation.

layer spanning from approximately 500 m to 1500 m. This unstable layer is topped by an inversion layer between 1500-2000 m, above which lies the free atmosphere. The initial horizontal wind is westward, characteristic of the subtropical trade winds. Constant surface fluxes of both sensible and latent heat are applied, along with prescribed large-scale subsidence. All surface boundary conditions are adopted from Siebesma et al. (2003) and remain constant for the duration of the run, including a sea surface pressure of 1015 hPa, a sea surface temperature of 300.4 K, and a surface air specific humidity of 22.45 g kg$^{-1}$. The

grid size and spacing also follow Siebesma et al. (2003), as listed in Table 1, and the case is run for six hours.

     Figure 5 shows vertical profiles of water vapor and cloud liquid water averaged over the last hour of the simulation, and their isotopic ratios averaged over each hour of the simulation. Cloud liquid water results for the BOMEX case from the multi-model LES ensemble (Siebesma et al., 2003) and MicroHH (van Heerwaarden et al., 2017; Bastak Duran et al., 2021) are also shown for context. The iPyCLES results are generally in good agreement with previous model simulations, but with larger values of

$q_l$ near the cloud top (1200–1500 m), suggestive of a small anvil. The isotopic ratios are reported in the standard $\delta$ notation, as introduced in eq. (39). The isotopic ratio $\delta_v^{18}$O decreases with increasing height, especially around the top of the cloud, and deviates only slightly from its initial state over the six-hour integration. Careful examination of the vertical profiles suggests three distinct regimes. Ratios of heavy isotopes in water vapor become more depleted between the surface and approximately 1000 m, corresponding to the sub-cloud layer and the lower portion of the cloud. By contrast, ratios of heavy isotopes in the

upper part of the cloud layer (1000–2000 m) become more enriched over time. These two trends can be explained by turbulent mixing within the cloud, which transports relatively enriched water upward and relatively depleted water downward. As water vapor and cloud liquid water are assumed to remain in isotopic equilibrium at all times, turbulent fluxes of both water vapor and cloud liquid water contribute to this effect. Isotopic ratios above the cloud layer are essentially unchanged over the duration of the simulation. The simulated profiles of deuterium excess in water vapor (d$_v$; Figure 5c) offer additional information on

the performance of the isotopic module. Values of d$_v$ increase over time, taking form as a series of steps that decrease in the



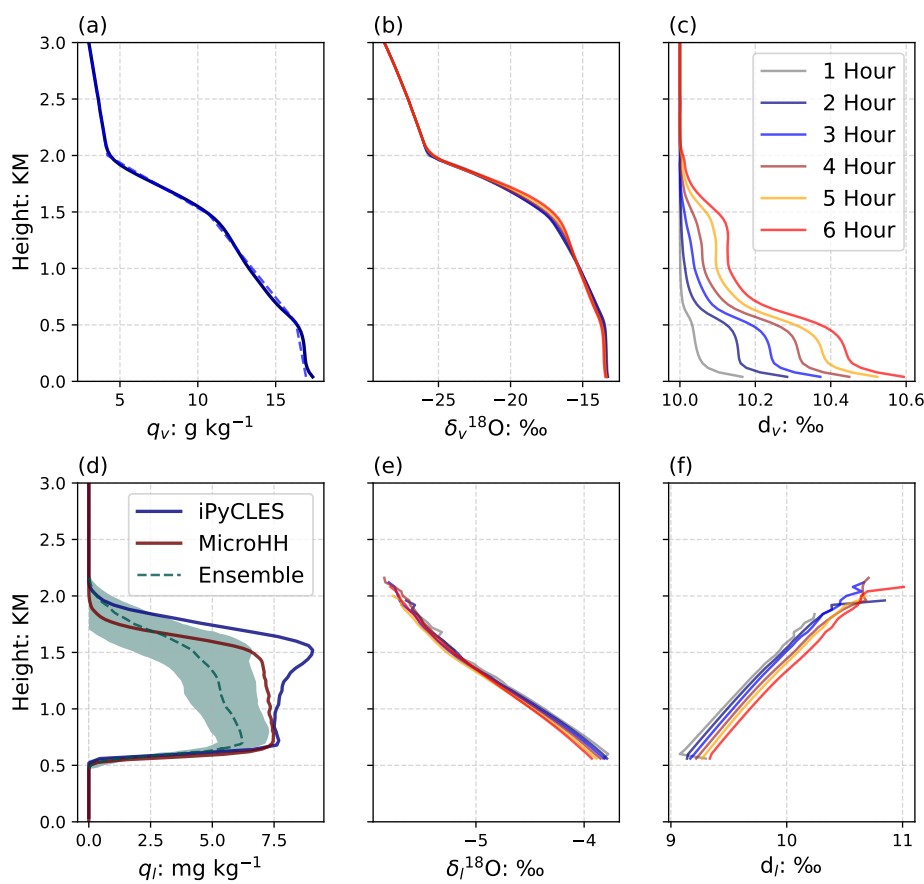

**Figure 5.** Vertical profiles of (a) domain-mean water vapor specific humidity over the last hour of the simulation and hourly-averaged (b) $\delta^{18}$O and (c) deuterium excess ($d$) in water vapor; (d) domain-mean specific cloud liquid water content over the last hour of the simulation and hourly-averaged (b) $\delta^{18}$O and (c) deuterium excess ($d$) in cloud liquid water from the BOMEX simulation. The dashed green line and shaded area in (c) show the LES ensemble mean and spread for the same case from Siebesma et al. (2003), and the dark red line in (c) shows a corresponding MicroHH simulation result (Bastak Duran et al., 2021).





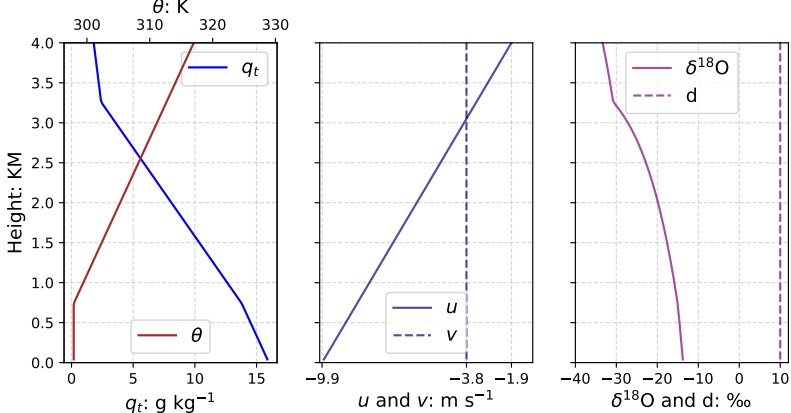

**Figure 6.** Initial profiles for the RICO case: (a) total specific water content ($q_t$) and potential temperature ($theta$); (b) eastward ($u$) and northward ($v$) components of the horizontal wind; and (c) the isotopic variables $\delta^{18}O$ and deuterium excess ($d$)

surface layer (0–50 m), at cloud base ($\sim$500 m), and in the anvil portion of the cloud (1500–2000 m). The increase in deuterium excess over time shows the influence of the surface source through the C-G model, where kinetic fractionation associated with unsaturated air over a smooth surface acts to increase d (Machavaram and Krishnamurthy, 1995; Benetti et al., 2014). Increases in deuterium excess with height in cloud liquid water are consistent with liquid–vapor isotopic equilibrium associated with the
vertical temperature decrease within the cloud, which drops from approximately 295 K at cloud base to 287 K at cloud top.

## 3.2   The RICO Case: Precipitating Subtropical Cumulus

To assess the added influence of microphysical processes on isotopic composition in warm stratocumulus regimes, particularly isotopic exchange between water vapor and rain droplets, we adopt a second test case based on observations collected during the RICO campaign (Rauber et al., 2007). This test case has been the focus of an LES intercomparison experiment reported
by van Zanten et al. (2011), and a modified version of the RICO case with an imposed diurnal cycle was used to test the performance of the isotope-enabled large-eddy warm cloud model ISOLESC (Wei et al., 2018). The RICO campaign was a comprehensive field study conducted in the northwest tropical Atlantic between late November 2004 and late January 2005, with a primary emphasis on analyzing the statistical properties of boundary layer clouds and precipitation associated with shallow convection (Gerber et al., 2008; Grabowski et al., 2011; Matheou et al., 2011; Lee et al., 2012a; Abel and Boutle,
2012; Zuidema et al., 2012; Arabas and Shima, 2013; Bruine et al., 2019).

Figure 6 shows initial conditions for thermodynamic, dynamic, and isotopic variables for the RICO case. The initial condition is cloud free ($q_v = q_t$) and under the influence of northeasterly trade winds. The settings closely resemble those employed in the LES intercomparison conducted by van Zanten et al. (2011), who used mean values derived from nearby radiosonde observations between 16 December 2004 and 8 January 2005 (Rauber et al., 2007). The simulation spans 20 hours to allow
adequate time for clouds and precipitation to develop.





Figure 7 illustrates the temporal evolution of the marine cloud layer and precipitation from the RICO case, including cloud cover (Fig. 7a), cloud liquid water path (LWP; Fig. 7b), and rain water path (RWP; Fig. 7c), along with the vertical profiles of cloud fraction (Fig. 7d), specific cloud liquid water content (Fig. 7e), and specific rain water content (Fig. 7f) averaged over hours 16–20 of the simulation. Reference values from a multi-model ensemble (van Zanten et al., 2011) and simulations

based on MicroHH (van Heerwaarden et al., 2017; Bastak Duran et al., 2021), a LES framework implemented in the ICON (ICOsahedral Nonhydrostatic) unified modeling system by Dipankar et al. (2015), are shown for the context where available. All models show a clear spin-up signal at the onset of the simulation, indicating the initial development of a turbulent but largely non-precipitating cloud layer (see also Wei et al., 2018). Cloud cover based on iPyCLES is slightly smaller than that in the multi-model ensemble. LWP shows good consistency with the ensemble and MicroHH, with all simulations showing a

gradual increase in LWP trend after the initial spike at spin-up. Variations in RWP are larger in iPyCLES than in the multi-model ensemble, with repeated rain events over the last 8–12 hours of the simulation. Surface rain rates associated with these events peak at around $1 \, \text{mm day}^{-1}$, consistent with observations collected during the RICO campaign (Stevens and Seifert, 2008). Vertical profiles of domain-mean cloud fraction and specific cloud liquid water content are likewise broadly comparable to the multi-model ensemble and MicroHH. Cloud fraction peaks near the cloud base at a value of approximately 0.06, with

strong consistency across the models. The overall shape of the cloud fraction profile is also similar across models, although iPyCLES produces larger cloud fractions in the upper part of the cloudy layer. Cloud liquid water increases from the cloud base at about 500 m to a peak near the cloud top. This peak is slightly larger in magnitude in iPyCLES than in the models reported by van Zanten et al. (2011) and slightly lower in altitude than that simulated by MicroHH (van Heerwaarden et al., 2017; Bastak Duran et al., 2021), but is again broadly consistent. The vertical profile of $q_r$ (Fig. 7f) (d) shows much larger

concentrations of rain water than either the multi-model ensemble or MicroHH within the cloud layer. Differences in $q_r$ are reduced in the sub-cloud layer, indicating that the iPyCLES simulation includes strong evaporation of falling rain in the unsaturated boundary layer.

Figure 8 shows vertical profiles of specific water contents and isotopic ratios in water vapor, cloud liquid water, and rain water from the RICO case. Relative to the initial state, the water vapor specific humidity (Fig. 8a) increases slightly near the

surface while the vertical gradient weakens in the cloud layer and sharpens above the cloud layer. The increase near the surface derives from the surface evaporation, the weaker gradient within the cloud from turbulent mixing, and the sharper gradient above the cloud from the imposed large-scale subsidence. Changes in the vertical profile of $\delta^{18}O$ (Fig. 8b) in water vapor show little variation in the sub-cloud layer from the surface to approximately 500 m, consistent with well-mixed conditions. By contrast, kinetic fractionation in the surface flux results in a distinct increase of deuterium excess in water vapor below the

cloud. Within the lower part of the cloud, around 500–1250 m, the value of $\delta_v^{18}O$ in water vapor decreases slightly during the model run. This change is accompanied by increases in $\delta_v^{18}O$ in the upper part of the cloud layer, around 1250–2000 m. These changes are caused by turbulent mixing of both water vapor and condensate, especially cloud liquid water (Fig. 9a-c). Water vapor is more enriched in heavy isotopes at lower altitudes, as is the liquid water that forms from it. Isotopic equilibration of cloud liquid water with water vapor after mixing further reinforces the mixing signal from water vapor alone. This mixing

signal can be even more clearly identified by plotting variations in water vapor specific humidity $q_v$ against the corresponding

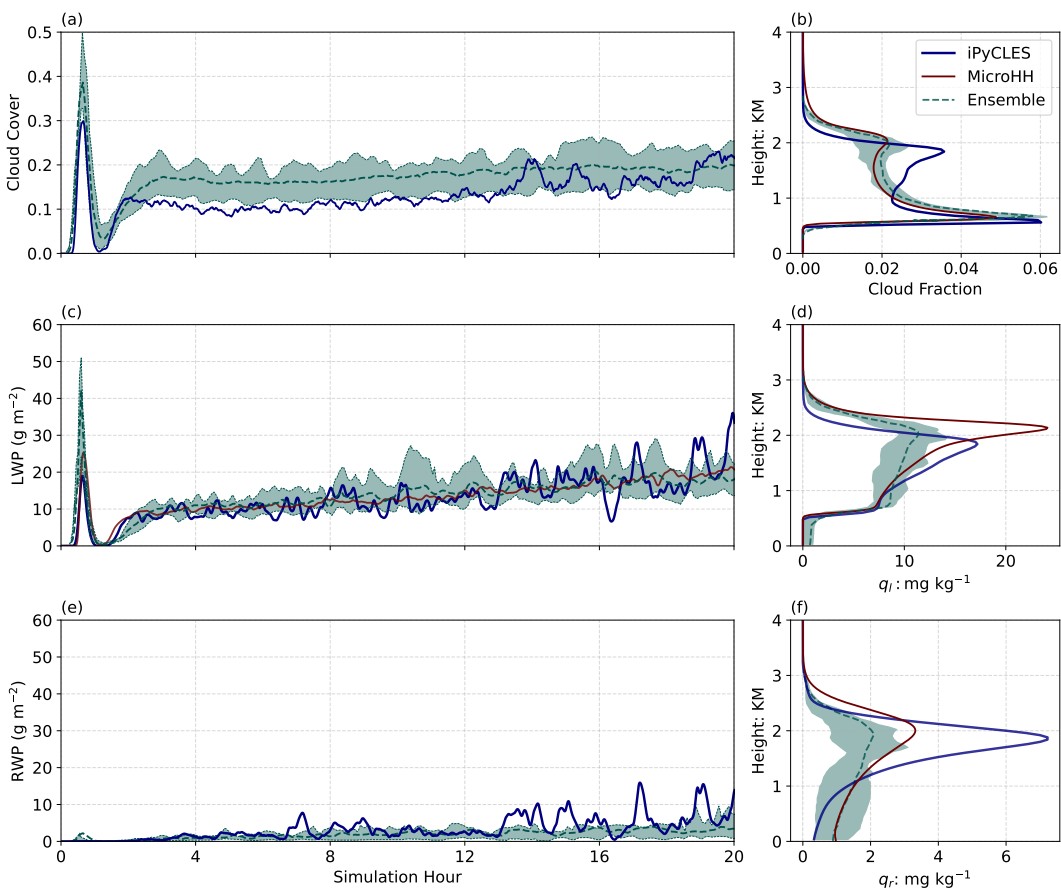

**Figure 7.** Time series of (a) cloud cover and vertically-integrated (c) liquid water path (LWP) and (e) rain water path (RWP) during the RICO case based on iPyCLES (dark blue). The right panels show vertical profiles of horizontal domain-averaged (b) cloud fraction, (d) specific cloud liquid water content $q_l$, and (f) specific rain water content $q_r$ averaged over the last four hours of the simulation. The green dashed line and shading in each panel indicate the multi-model ensemble mean and spread from an earlier RICO intercomparison (van Zanten et al., 2011), while the dark red line shows a corresponding MicroHH simulation result (Bastak Duran et al., 2021).




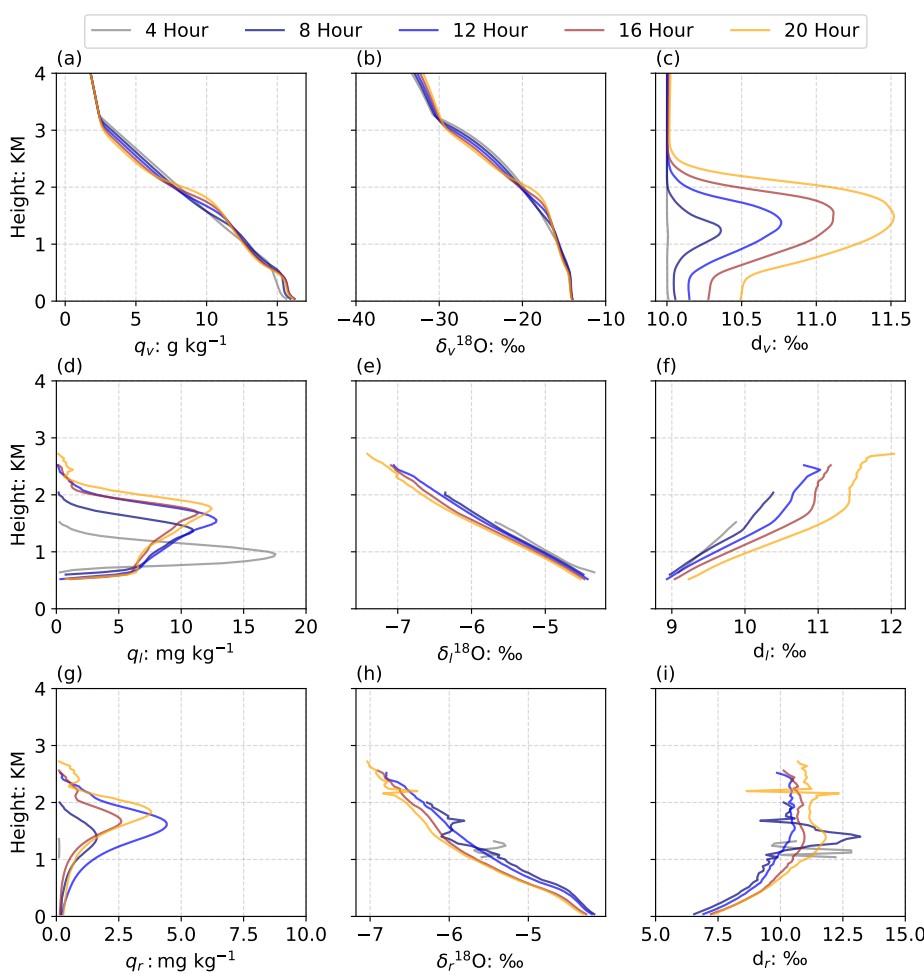

**Figure 8.** Vertical profiles of hourly-averaged (a) water vapor specific humidity, (b) $\delta^{18}$O in water vapor, and (c) deuterium excess d in water vapor. Corresponding hourly-averaged specific water contents and isotopic compositions are also shown for (d-f) cloud liquid water and (g-i) rain. Profiles are averaged over the hour prior to the simulation time marked in the legend.





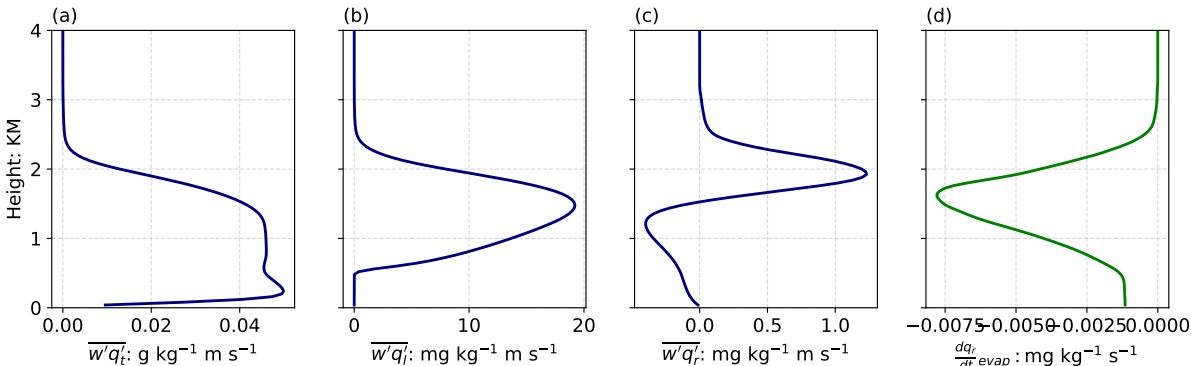

**Figure 9.** Domain-mean vertical turbulent fluxes from the RICO case for (a) total specific water content $q_t$, (b) specific cloud liquid water content $q_l$, and (c) specific rain water content $q_r$, along with (d) the evaporation tendency for specific rain water content: $\frac{dq_r}{dt}_{\text{evp}}$. All profiles are averaged from hour 4 to hour 20 of the simulation.

isotopic ratio $\delta_v^{18}$O (Fig. 10) and comparing these distributions to theoretical models. Water vapor specific humidities within the cloud layer range from about $15\,\mathrm{g\,kg^{-1}}$ at cloud base to about $6\,\mathrm{g\,kg^{-1}}$ at cloud top (Fig. 8a). Isotopic ratios within this range of specific humidities show a distinct bend away from Rayleigh distillation (i.e. the initial condition) and toward the theoretical mixing line.

Larger increases in $\mathrm{d}_v$ (Fig. 8c) in the cloud layer relative to the sub-cloud layer illustrate the effect of kinetic fractionation during rain evaporation. In comparison to equilibrium fractionation, differential diffusivity between HDO and $\mathrm{H_2^{18}O}$ results in the proportion of HDO that evaporates into the surrounding air during incomplete evaporation of raindrops exceeding that of $\mathrm{H_2^{18}O}$. This effect leads to an enrichment of deuterium relative to $^{18}$O and a corresponding increase in $\mathrm{d}_v$. The pronounced peak in rain evaporation within the cloud layer (Fig. 9d) supports this interpretation. Decreases in $\delta_v^{18}$O above the cloud layer
indicate the influence of imposed large-scale subsidence, while slight enrichment above 3000 m shows that cloud detrainment occasionally mixes upward into the deeper free troposphere (see also deviation toward the mixing line at small specific humidities in Fig. 10). Although this mixing is weak, relatively small water vapor specific humidity values at this altitude allow the isotopic signal to emerge (Fig. 8b-c) despite no evident change in the specific humidity (Fig. 8a). This isotopic sensitivity is a leading motivation for using water isotopes to evaluate sources of humidity in the subtropical free troposphere (e.g. Noone
et al., 2011; Galewsky et al., 2011).

    As in the BOMEX case, the evolution of isotopic ratios in cloud water ($\delta_l^{18}$O and $\mathrm{d}_l$; Figure 8e-f) closely follows that in water vapor, with the increased sensitivity in $\mathrm{d}_l$ again consistent with liquid–vapor equilibrium at the range of temperatures within the cloud. These results are consistent with the simulation conducted by Wei et al. (2018), although their simulation produced larger signals due to an imposed diurnal cycle and the inclusion of a transpiration surface source. The largest concentrations
of specific rain water content are located just below 2000 m through most of the run, in the upper part of the cloud layer. The isotopic ratio in rain ($\delta_r^{18}$O; Fig. 8h) exhibits a similar distribution to that in cloud liquid water, but is slightly enriched in





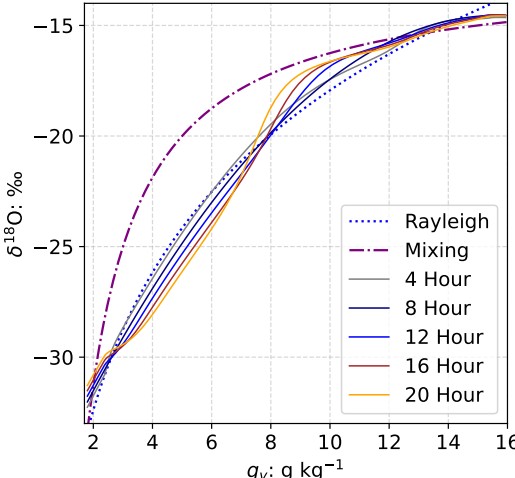

**Figure 10.** Joint variations in domain-mean water vapor specific humidity $q_v$ and $\delta_v^{18}$O during the RICO simulation. Theoretical curves for Rayleigh distillation and an air mass mixing model (Noone, 2012) are shown for reference. The plotted curves correspond to the profiles shown in Fig. 8a-b. The Rayleigh distillation curve corresponds to the initial conditions prescribed in the model run.

the heavy isotope relative to cloud liquid water. Several processes within the cloud layer impact the isotopic composition of rain droplets, including autoconversion and accretion of $q_l^{iso}$, equilibration with water vapor, and kinetic fractionation when rain evaporates in unsaturated grid cells. The deuterium excess of rain drops within the cloud layer ($d_r$; Fig. 8h) remains very

close to 10‰ throughout the simulation. Smaller values of $d_r$ in the sub-cloud layer can be attributed to preferential loss of deuterium relative to $^{18}$O as falling raindrops evaporate into the surrounding air.

### 3.3 Mix-Phase Arctic Cloud: ISDAC Case

To assess the performance of iPyCLES using mixed-phase microphysics, we conduct a third test case focusing on a mixed-phase stratocumulus cloud observed on 26 April 2008 during the Indirect and Semi-Direct Aerosol Campaign (ISDAC; Mc-

Farquhar et al., 2011). Two simulations are conducted for this case to evaluate differences between the one-moment Kaul15 and two-moment SB06 mixed-phase microphysics and their impacts on the simulated isotopic tracers. The observations were collected north of Barrow, Alaska, for a mixed-phase cloud layer that persisted for approximately 15 hours. The cloud layer was located over sea ice, was largely decoupled from the surface, and was capped by large-scale subsidence, rendering it highly suitable for analysis in idealized LES models. In addition to the observations, an intercomparison study of 12 LES models based on this case has been reported by Ovchinnikov et al. (2014). The ISDAC case was previously implemented in PyCLES

by Zhang et al. (2020) using the Kaul15 scheme, with initial conditions as summarized in Figure 11. Pronounced inversions of potential temperature and moisture are prescribed at a height of 850 m, marking the lower boundary of the free atmosphere. The initial profile of $\delta^{18}$O in water vapor is again calculated based on Rayleigh distillation but from a more depleted source than in the subtropical cases, with $\delta$D set assuming a deuterium excess of 10‰. Surface and large-scale boundary conditions



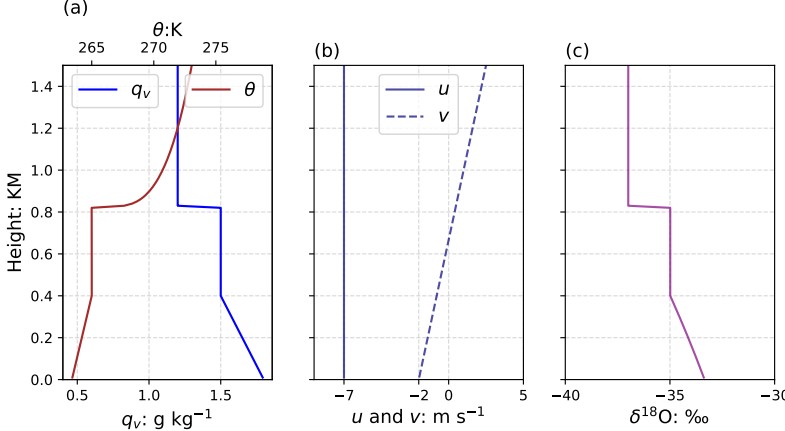

**Figure 11.** Initial profiles for the ISDAC case: (a) water vapor specific humidity ($q_v$) and potential temperature ($\theta$), (b) eastward ($u$) and northward ($v$) components of the horizontal wind, and (c) the isotopic ratio $\delta^{18}$O. The deuterium excess is set to be 10‰ as in previous cases.

follow the guidelines outlined by Ovchinnikov et al. (2014), with sensible and latent heat fluxes set to zero due to the limited interaction between the surface and the cloud layer. Large-scale subsidence is applied as an external potential temperature advection along with nudging for momentum, temperature, and moisture at each level. The ISDAC simulations are run for 8 hours.

Figure 12 shows time series of LWP, IWP, and snow water path (SWP) from both ISDAC model runs. Both simulations show
a spin-up signal in the initial half-hour followed by continuous growth of the cloud layer. The first hour of each simulation is disregarded in the following analysis. Both schemes produce LWPs between 10 and 40 g m$^{-2}$, with the Kaul15 scheme producing the larger value. The difference in LWP between the two simulations increases gradually with time, peaking at around 11 g m$^{-2}$. A qualitatively similar difference is found for IWP, with the Kaul15 result stabilizing at 6 g m$^{-2}$, about three times larger than IWP based on SB06. LWP and IWP from both runs are broadly consistent with previous model results
and field observations, with flight measurements suggesting LWP ranging from 3.7 to 24 g m$^{-2}$ and IWP ranging from 3.1 to 11 g m$^{-2}$ (McFarquhar et al., 2011; Zhang et al., 2020). The time series of SWP (Fig. 12c) also shows a striking difference between the two schemes, with SB06 producing more snow despite having smaller LWP and IWP.

Figure 13 shows domain-mean vertical profiles of key thermodynamic variables averaged over hours 1–8 of the simulation, along with observations collected on flight 31 of the ISDAC field campaign. Due to limitations in distinguishing ice and snow
in the observations, comparisons are based on the sum $q_i + q_s$. A well-mixed boundary layer is evident in both the cloud and sub-cloud layers, driven by radiative cooling of the mixed-phase cloud top. In particular, the horizontal domain-mean profiles of $q_t$ and the liquid–ice potential temperature $\theta_{li}$ (Tripoli and Cotton, 1981) become increasingly well-mixed during the simulation relative to the initial conditions, indicating vigorous turbulent mixing. The vertical structure of these variables





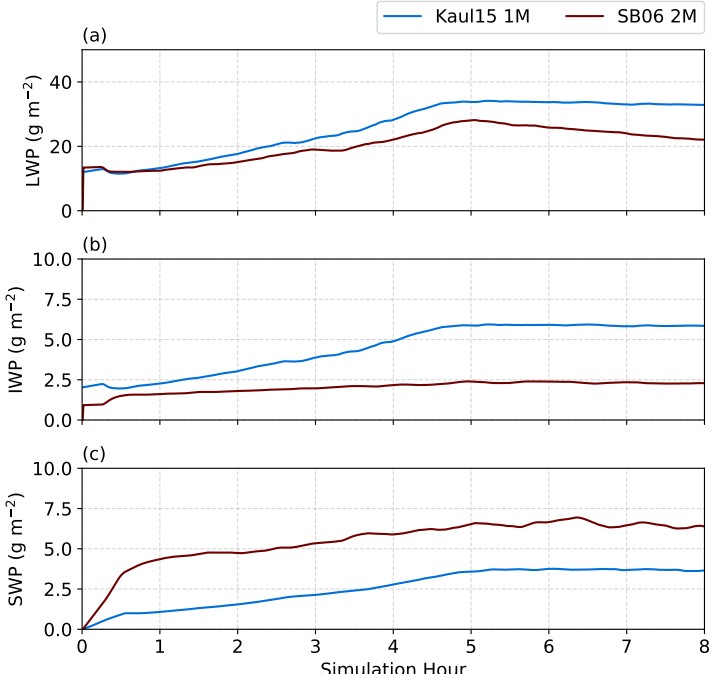

**Figure 12.** Time series of (a) cloud liquid water path (LWP), (b) cloud ice water path (IWP), and (c) snow water path (SWP) from the 8-hour ISDAC run. Results using both the Kaul15 one-moment (blue) and SB06 two-moment (red) microphysics are presented for comparison.

is consistent between the two simulations and in good agreement with observations, although the additional snow produced by

the two-moment scheme results in a slightly drier boundary layer relative to the one-moment scheme.

Within the mixed-phase cloud layer, both schemes overestimate specific cloud liquid water content compared to observations (Fig. 13c). Liquid water is concentrated near the top of the cloud in both simulations, consistent with the application of saturation adjustment for vapor–liquid equilibrium in both schemes, with a shallower liquid layer and a slightly smaller peak value in the simulation using the two-moment scheme relative to that using the one-moment scheme. This approach is suitable

for liquid clouds due to the short relaxation time between water vapor and cloud droplets, and helps to keep computational costs and biases small. However, the generation of ice particles in mixed-phase clouds involves complex interactions among vapor, cloud liquid, and cloud ice, and saturation adjustment can lead to substantial biases relative to observations. Unlike the Kaul15 scheme, which uses saturation adjustment for both liquid and ice, the SB06 scheme calculates ice concentrations based on parameterized nucleation. This approach provides better agreement with observed ice particle concentrations in the

mixed-phase cloud layer in the ISDAC case (Fig. 13d). It is also important to note that $q_i$ is a diagnosed variable in the Kaul15 scheme but a prognostic variable in the SB06 scheme. Some of the ice concentration in the sub-cloud layer may therefore result from numerical diffusion in addition to the effect of settling. With respect to snowfall, both schemes underestimate the





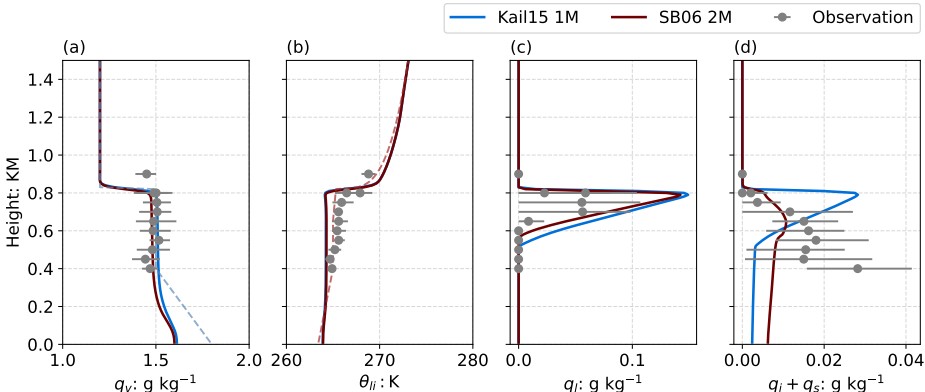

**Figure 13.** Vertical profiles of horizontal domain-mean (a) total specific water content $q_t$, (b)liquid and ice potential temperature $\theta_{li}$, (c) specific cloud liquid water content $q_l$, and (d) total specific water content of cloud ice and snow $q_i + q_s$ from the simulations using the Kaul15 (blue) and SB06 (red) mixed-phase microphysics. All profiles are averaged over hours 1–8 of the simulation. The initial conditions are shown as dashed lines in (a) and (b) for context. The grey dots show field measurements collected during flight 31 of ISDAC campaign on 27 April 2008, with uncertainty bars encompassing the 15–85% range of measurements collected at 50-m intervals from 400–900 m height (McFarquhar et al., 2011, their Fig. 14).

observed snow and ice concentrations at lower altitudes (Fig. 13d). However, the SB06 scheme demonstrates stronger snowfall and a more rapid onset of precipitation compared to Kaul15 (Fig. 12c), along with less sublimation in the unsaturated sub-cloud layer. These differences arise directly from details of the microphysical parameterizations.

Figure 14 compares the evolution of $\delta^{18}$O in each water species between the two simulations. Overall, the relative values of $\delta^{18}$O among the water species evolve as expected during the run, with liquid and ice species each progressively more enriched in heavy isotopes than water vapor. Heavy isotope ratios in water vapor decrease with increasing altitude from the surface, particularly early in the run (Fig. 14a). Strong mixing in the sub-cloud layer produces an increasingly well-mixed profile as the run evolves, but the decrease of $\delta^{18}$O with height in the liquid cloud layer is maintained (Fig. 14m). The profiles also gradually become more depleted in $^{18}$O as snow is produced and falls out. This change is more pronounced in the SB06 simulation, consistent with its greater production of snow. Although snow and ice are initially more enriched in heavy isotopes in the Kaul15 run relative to the SB06 run (Fig. 14c-d), progressive depletion in Kaul15 first reverses this difference and then leads to an increasing separation of $\delta^{18}$O in ice and snow between the two runs. By the end of the simulations, snowfall $\delta_s^{18}$O is approximately 2‰ more enriched in the SB06 run (Fig. 14p). Surface snowfall over time would thus show significant differences in the evolution of isotopic composition between these two scenarios, with a declining trend based on Kaul15 but a fairly steady ratio based on SB06.

Figure 15 shows the evolution of vertical profiles of deuterium excess ($d$) in all water species during the two runs. The SB06 run produces a substantial increase in $d_v$ (~2‰ over 8 hours) in boundary layer water vapor that is not reproduced in the Kaul15 run. This increase results from kinetic fractionation during the deposition of water vapor to ice and snow. The



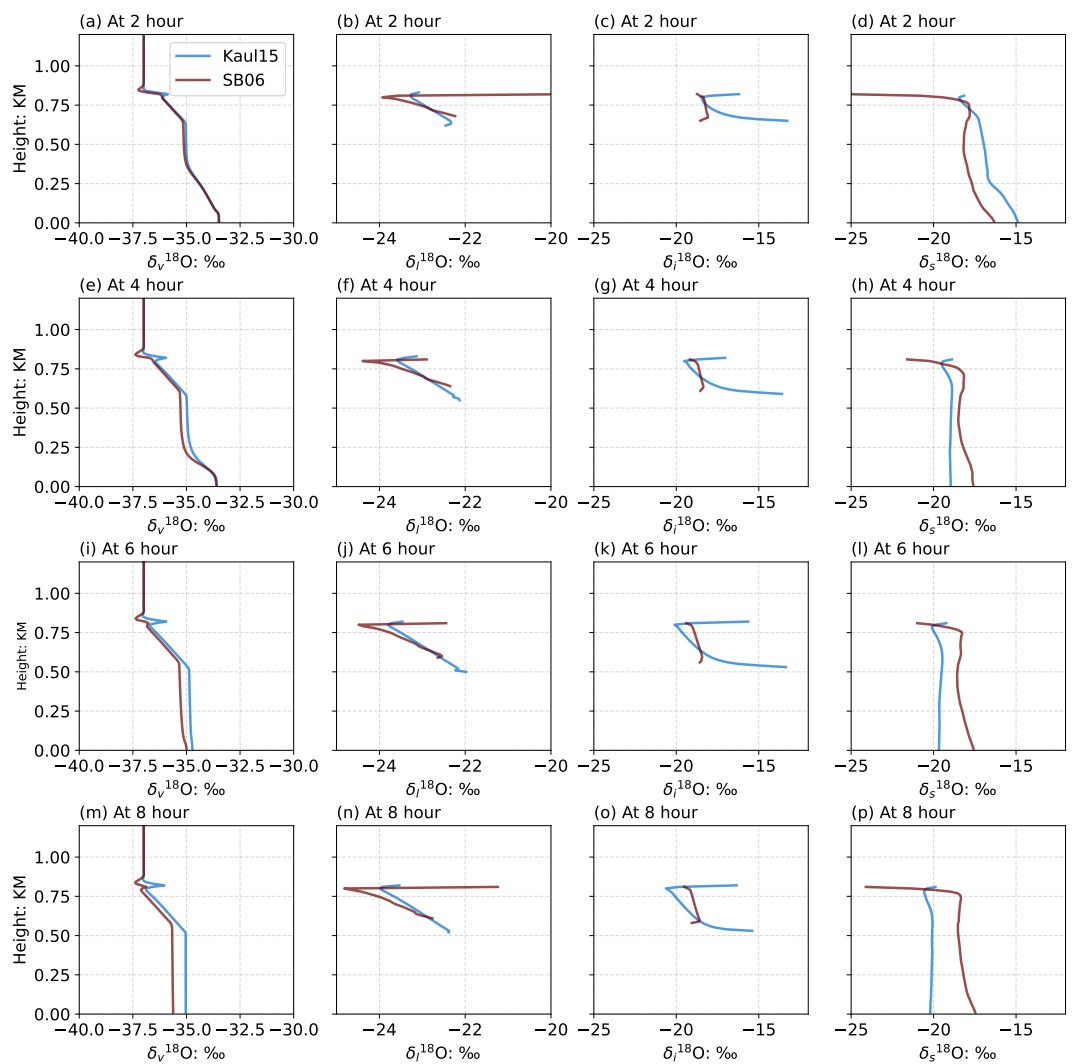

**Figure 14.** Horizontal domain-mean vertical profiles of $\delta^{18}O$ in water vapor (first column), cloud liquid water (second column), cloud ice (third column), and snow (fourth column) simulated using iPyCLES with the Kaul15 microphysics (blue) and the SB06 microphysics (red). Profiles are 1-hour averages preceding hours 2, 4, 6, and 8 of each simulation.





preference for the isotopically heavier $H_2^{18}O$ molecules to condense is amplified during deposition, leading to a larger residual concentration of HDO in water vapor. The isotopic composition of cloud liquid water is determined by equilibration with the surrounding water vapor, so that changes in $d_l$ follow those in $d_v$. More striking differences are found in the evolution of deuterium excess in ice and snow particles. Based on Kaul15, values of $d_i$ and $d_s$ gradually increase from large negative

values early in the run (Fig. 15c-d) to approach zero by the end of the run (Fig. 15o-p). This increase is in stark contrast to very small changes in $d_i$ and $d_s$ simulated in the SB06 run. The cloud ice content is diagnosed in Kaul15 but can be recirculated by boundary layer mixing and in the SB06 scheme (Fig. 15g-i), which may account for the more stable isotopic composition and snow in the run using SB06. Snow primarily originates from the transformation of cloud ice particles and the aggregation of small ice and liquid droplets, so that the isotopic ratio $\delta_s$ closely resembles $\delta_i$ within the mixed-phase cloud layer in both

schemes. Differences are larger below the cloud, where the Kaul15 scheme produces values that are roughly constant in height but SB06 indicates a reduction in $d_s$ as it approaches the surface. This difference may indicate a relationship between snow isotopic content and fall rate in SB06 that emerges from the sharp gradient of $d_l$ in the liquid layer (e.g. Fig. 15n), which is not present in Kaul15. Future sensitivity tests will be needed to fully diagnose and understand these differences in the isotopic results.

The profiles of both $\delta_v^{18}O$ and $d_v$ in water vapor show noticeable kinks near the inversion level (around 800–850 m) in both simulations (left columns of Figs. 14–15). The SB06 simulation result shows a particularly sharp peak in $d_v$ at this altitude that becomes increasingly pronounced over time (Fig. 15a,m) but is not evident in vertical profiles of domain-mean $d_v$ based on the Kaul15 scheme. To further investigate this distinctive feature, Figure 16 shows vertical profiles of horizontal domain-mean $d_v$ together with the vertical turbulent flux and the tendency of total water due to this flux. Given the strong influence of

kinetic fractionation on $d_v$, the net deposition and sublimation tendencies of ice and snow for SB06 (snow only for Kaul15), the turbulent flux of cloud liquid water for SB06, and the evaporation and condensation tendency of cloud liquid water for SB06 are also shown. Cloud condensate transport and tendency terms are not available for the Kaul15 run because cloud ice and cloud liquid water are diagnostic rather than prognostic. The altitude of the peak in $d_v$ is located at 820 m, the height of the inversion layer in the initial settings. The turbulent tendency also exhibits a clear peak at this altitude (Fig. 16c), consistent

with strong turbulent mixing at the cloud top due to unstable thermodynamic conditions resulting from cloud-top radiative cooling. The turbulent flux of total water (Fig. 16b) further shows that the large positive tendency of total water at the cloud top originates from the mixed-phase layer below, where strong kinetic fractionation during vapor deposition to snow and ice particles leads to enrichment of $d_v$. However, whereas the turbulent $q_t$ tendency is reproduced in both the SB06 and Kaul15 schemes, the prominent spike in $d_v$ appears only in the results based on SB06, indicating that other factors are involved.

Although sublimation of ice and snow also a source of water vapor above the cloud top (Fig. 16d), this source cannot account for the sharp peak in $d_v$ because ice and snow also carry smaller values of deuterium excess than water vapor and iPyCLES does not consider isotopic fractionation during sublimation.

The turbulent flux and evaporation tendency of cloud liquid water both exhibit peaks near the cloud top (Fig. 16e-f) and could serve as additional influences on $d_v$. Isotopic exchange between cloud liquid water and water vapor is treated as equilibrium

fractionation. The deuterium excess of liquid water in equilibrium with water vapor also exhibits a temperature dependence



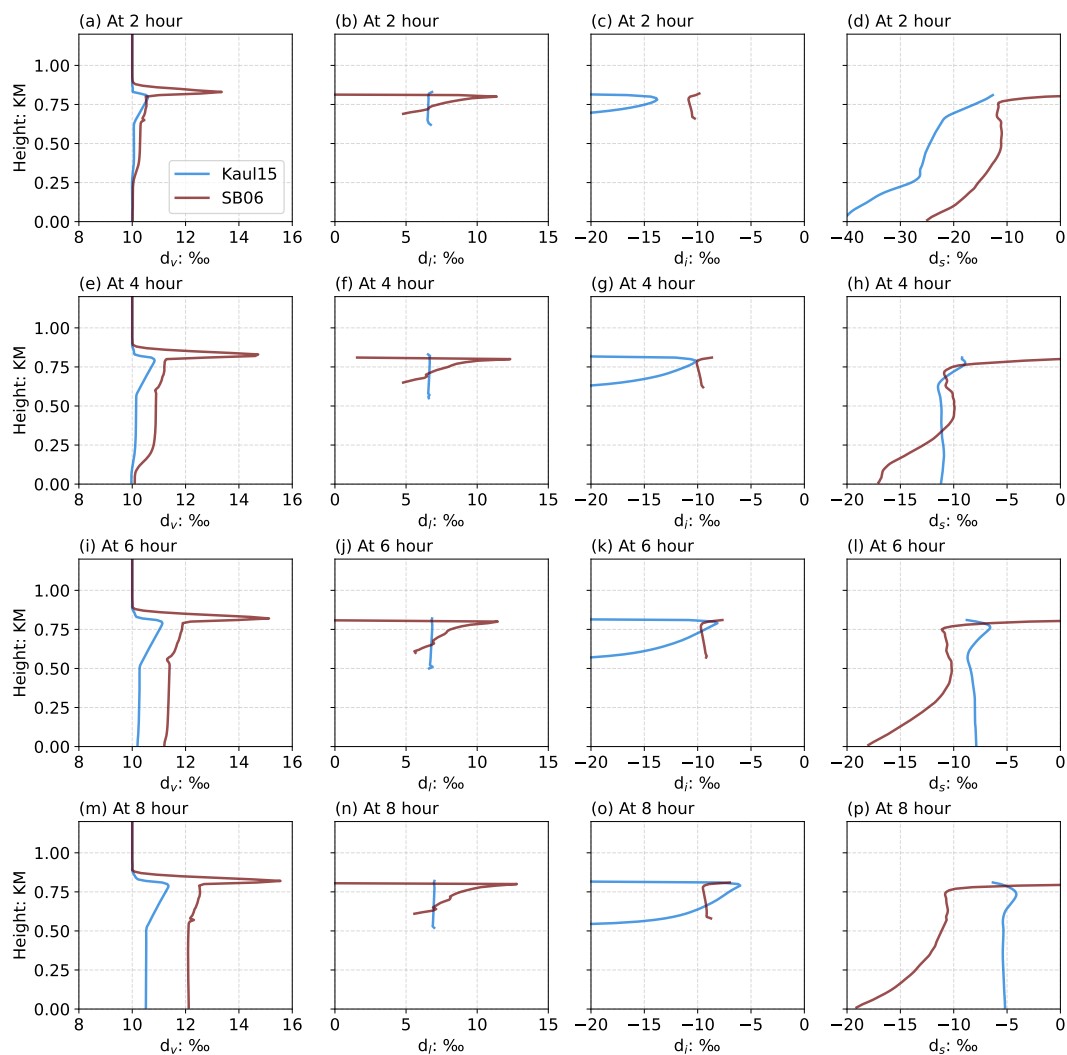

**Figure 15.** Horizontal domain-mean vertical profiles of deuterium excess ($d$) in water vapor (first column), cloud liquid water (second column), cloud ice (third column), and snow (fourth column) simulated using iPyCLES with the Kaul15 microphysics (blue) and the SB06 microphysics (red). Profiles are 1-hour averages preceding hours 2, 4, 6, and 8 of each simulation.



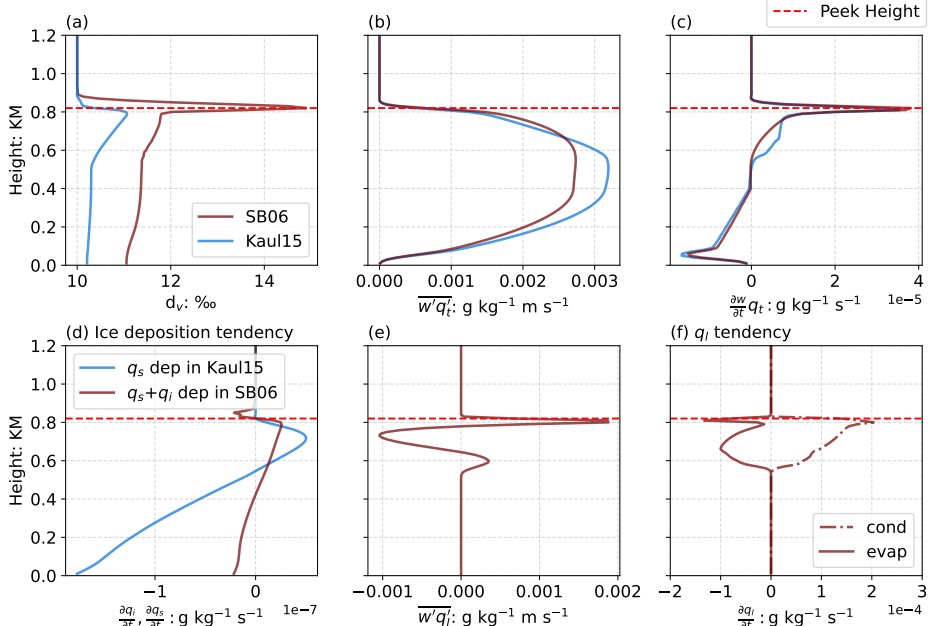

**Figure 16.** Vertical profiles of horizontal domain mean of diagnosed variables to explain the isotopic peek at cloud topic within SB06 scheme, with the compression from Kaul15 scheme: (a) profiles of deuterium excess of water vapor $d_v$; (b) vertical flux of total specific water content $q_t$; (c) vertical flux tendency of $q_t$; (d) deposition and sublimation tendency of specific cloud ice water content $q_i$; (e) vertical flux tendency of $q_l$; (f) condensation(cond) and evaporation(evap) tendency of specific cloud liquid water content $q_l$. The altitude of the isotopic peak is plotted by a dashed red line with 820 m height. The profiles from Kaul15 scheme are plotted in purple. Ice deposition/sublimation and cloud liquid evaporation/condensation in Kaul15 scheme are not included because of the diagnosed $q_i$ and $q_l$ within the mixed-phase saturation adjustment.

(see, e.g., Fig. 5f). The direction of this temperature dependence means that cloud liquid water formed at a colder temperature (i.e. in the mixed-phase cloud layer below the inversion) that then equilibrates with water vapor at a warmer temperature (i.e. above the inversion) will tend to shift excess deuterium toward the warmer temperature. The peak in $d_v$ produced by the SB06 scheme may therefore emerge as a special feature of isotopic equilibration between water vapor and cloud liquid water

occurring within the sharp vertical temperature gradient of the cloud-top inversion. It is important to note that this effect may be overestimated by iPyCLES-SB06 in its current form, as partial evaporation is assumed to shrink all liquid water drops by a given fraction rather than evaporating a fraction of the drops, thereby maximizing fractionation. However, the point remains that the ability to transport and mix prognostic cloud liquid water and ice and the elimination of the saturation adjustment procedure produce a potentially observable signature in the isotopic composition of water vapor near the top of the Arctic

boundary layer.

    Additional differences between the schemes include the vertical structure of deposition to snow and ice, which is shifted noticeably upward toward the top of the liquid cloud layer in SB06 relative to Kaul15 (Fig. 16d). The strongest kinetic fraction-



ation effects should occur close to the cloud top due to cold temperatures and large values of $S_i$ (eq. (29)). This difference may help to account for the remarkable enrichment of $\mathrm{d}_v$ at the inversion height, as well as the vertical profile of deuterium excess

in snow in SB06, provided that snow hydrometeor masses and fall rates are likewise correlated with the strength of kinetic fractionation. Strong sublimation of falling snow in the sub-cloud layer in the Kaul15 simulation (Fig. 16d) and subsequent recycling of that water vapor may also contribute to maintaining a more stable isotopic composition of water vapor in time relative to the SB06 result (left columns of Figs. 14–15). The net effect of these differences is that the temporal evolution of isotopic composition in snow reaching the surface, and particularly the deuterium excess, is remarkably different between the

microphysical parameterizations in ways that could be tested by comparison with observations.

## 4 Summary and Discussion

Simulations of isotopic evolution in precipitating and non-precipitating cumulus-topped marine boundary layers based on iPyCLES were evaluated using the BOMEX and RICO test cases. Overall, iPyCLES produces time series and vertical profiles of cloud liquid water and rain that agree well with multi-model ensembles produced for previous intercomparison stud-

ies (Siebesma et al., 2003; van Zanten et al., 2011; Bastak Duran et al., 2021). The vertical distribution of $\delta^{18}O$ in boundary layer water vapor is influenced primarily by vertical motion and equilibrium fractionation. Turbulent mixing in the sub-cloud layer and cloud-top entrainment instability both contribute to transporting water vapor enriched in heavy isotopes upward into and through the cloud layer. Notably, the isotopic profiles in the RICO case exhibit more pronounced vertical mixing than those in the BOMEX case, consistent with a more intense convective environment and latent heating from the production of

rain production. Equilibration between water vapor and cloud liquid water also helps to shape the vertical profile of $\delta^{18}O$ in water vapor, as equilibration with cloud droplets formed at lower altitudes enriches the upper cloud layer and vice versa. The isotopic content of water vapor is also influenced by kinetic fractionation in the surface evaporative flux, as represented by an implementation of the C-G model. Kinetic fractionation during partial evaporation of raindrops in unsaturated grid cells also influences values of deuterium excess in water vapor and rain in the RICO case, leading to increases in $\mathrm{d}_v$ at the maximum

evaporation level (about 1500 m) and decreases in $\mathrm{d}_r$ in the sub-cloud layer. These signals could be observed by measurements of isotopic ratios in water vapor in the boundary layer and lower free troposphere, as well as in the isotopic evolution of surface rainfall.

We further evaluate the evolution of water isotopes in an Arctic mixed-phase cloud layer using the ISDAC case (Ovchinnikov et al., 2014). Two runs were conducted to evaluate differences in the results between the Kaul15 one-moment and SB06

two-moment mixed-phase microphysical schemes. Comparison of the two simulations shows that the SB06 scheme produces smaller cloud liquid and ice water paths relative to Kaul15, but larger snow water path. This difference is directly related to details of the microphysical parameterization, especially the explicit representation of of cloud liquid activation and cloud ice nucleation in the SB06 scheme. The mixed-phase saturation adjustment method used in Kaul15, which assumes all water vapor above the saturation threshold instantaneously becomes cloud liquid water or cloud ice, tends to overestimate mixed-phase

cloud water content and underestimate the concentrations of cloud ice below the liquid layer compared to observations. The





SB06 scheme, which eliminates mixed-phase saturation adjustment, simulates concentrations of $q_i + q_s$ much more consistent with measurements collected during flight 31 of the ISDAC experiment. These results confirm the reliability of the PyCLES model for simulating Arctic mixed-phase clouds, as previously shown by Zhang et al. (2020), and highlight the considerable advantages of the two-moment SB06 scheme in simulating mixed-phase clouds. These advantages are further accentuated by

the isotopic tracers, as the vertical distribution of isotopes is predominantly shaped by coupling with the microphysics as the cloud layer in this simulation is largely decoupled from the surface. Fluctuations in $\delta_v^{18}$O and $\mathrm{d}_v$ over time are predominantly driven by removal of condensed water species, as well as mixing and subsequent equilibration between cloud liquid droplets and water vapor. In particular, the simulation using the SB06 two-moment microphysics produces a sharp peak in deuterium excess at the cloud top, which appears to derive from both kinetic fractionation during deposition of water vapor to ice and

snow (which leaves water vapor in the cloud layer more enriched in deuterium) and liquid–vapor equilibration within the sharp temperature gradient of the cloud top inversion (which acts to shift excess deuterium upward). The magnitude of this signal could be linked to the intensity of cloud top mixing, liquid–ice partitioning, and other characteristics of mixed-phase clouds and should be evaluated in further modeling studies and measurement campaigns. The evolution of the isotopic tracers in the ISDAC simulations underscores the advantages of two-moment microphysics for simulating isotopic evolution in mixed-phase

clouds, along with the disadvantages of mixed-phase saturation adjustment. These two changes support a better representation of the complex isotopic exchange among vapor, liquid, and ice phase reservoirs that occurs during the Wegener-Bergeron-Findeisen process in mixed-phase clouds. Moreover, the addition of a comprehensive mixed-phase microphysics scheme to the existing advantages of PyCLES, namely the use of WENO numerics and an energetically-consistent anelastic framework, supports expansion to additional applications, including deep convective clouds.

One significant limitation of this study is our use of Rayleigh distillation and the meteoric water line to set initial conditions for the isotopic tracers. This choice was made in part for simplicity, so that signals emerging over the duration of the simulation could be clearly compared to the initial settings, and in part by a lack of systematic observations of dual isotopes in water vapor vertical profiles. Certain assumptions in the isotope module also influence our simulated isotopic signals, such as the assumption that no fractionation occurs during the sublimation of ice to water vapor (Blossey et al., 2010). The effects of

these assumptions will be systematically evaluated using targeted sensitivity tests. The simplified parameterization of cloud droplet activation and ice nucleation, which assumes constant CCN and IN as input, presents another limitation that we intend to address in future development.

## 5 Conclusions

Motivated by the two key challenges in microphysical modeling outlined by Morrison et al. (2020), we introduce the new
isotope-enabled large eddy simulation model iPyCLES and test its performance in three widely used simulation cases. By incorporating the water isotopologues $H_2^{18}$O and HDO as passive water tracers, iPyCLES simulates the evolution of isotopic ratios in the atmospheric hydrological cycle under the influences of dynamics, thermodynamics, and cloud microphysics at fine horizontal and vertical scales. The further introduction of a two-moment mixed-phase microphysical scheme enables high-





resolution simulations of isotopic ratios in Arctic boundary layer clouds and reveals distinct isotopic signatures of turbulent mixing and microphysics in these cloud layers. The primary advantages of iPyCLES include:

– Two-moment ice phase cloud microphysics based on the SB06 scheme have been added to the existing warm-phase cloud microphysical parameterization to model the formation and evolution of mixed-phase clouds. This addresses several limitations of the one-moment microphysics scheme in simulating both the life cycle of mixed-phase cloud layers and the evolution of isotopic ratios within these layers.

– Isotopic tracers have been integrated into surface fluxes as well as the warm-phase, one-moment mixed-phase, and two-moment mixed-phase schemes, enabling iPyCLES to simulate a wide range of boundary layer conditions from subtropical shallow cumulus to Arctic mixed-phase clouds. Added to its energetically-consistent anelastic dynamics, the introduction of two-moment mixed-phase microphysics makes iPyCLES theoretically suitable for simulating isotopic evolution in deep convective clouds as well.

– Equilibrium and kinetic isotopic fractionation have been incorporated into all phase changes in iPyCLES. Three test cases have demonstrated the successful simulation of isotopic fractionation across water species in various boundary-layer environments.

– The mixed-phase saturation adjustment procedure used in the base model PyCLES has been replaced by parameterized nucleation of ice particles. This approach enables the representation of strong fractionation during the Wegener-Bergeron-Findeisen process by calculating supersaturation separately over liquid and ice rather than relying on a uniform pre-computed saturation vapor pressure, and allows vertical transport of cloud condensate to feed back on the isotopic composition of water vapor.

In conclusion, iPyCLES provides an isotope-enabled LES framework for studying atmospheric water cycling in clouds and boundary layers. The model addresses many of the limitations associated with coarse resolution and parameterized physics in large-scale isotope-enabled models such as global and regional climate models. By integrating isotopic tracers in an LES model with a comprehensive two-moment mixed-phase microphysics, iPyCLES can be used to conduct idealized experiments to better link the wealth of information provided by isotopic observations to improved understanding of physical processes and stronger observational constraints on parameterized physics in large-scale models. The amount and quality of isotopic observations have expanded substantially in recent years with the advent of new in situ and remote sensing-based measurement techniques, as well as the increasing inclusion of instrumentation for measuring isotope ratios in dedicated field campaigns. These observations reveal the distribution and variability of isotopes within specific domains but are difficult to integrate into a continuous and holistic vertical framework, limiting their applicability. Our objective in developing the iPyCLES model is to provide a tool that can help to fill this gap. The model, which uses strong stability-preserving WENO numerics and an anelastic LES framework, can be applied across a wide range of resolutions, from replicating cloud chamber experiments to simulating deep convective clouds. It could therefore help to enrich or supplement laboratory experiments, such as evaluating changes



in kinetic isotopic fractionation in mixed-phase clouds under varying concentrations of types of aerosol, or to test theoretical hypotheses regarding the isotopic imprint of mixed-phase layer depth in convective clouds, as proposed by Bolot et al. (2013).

*Code and data availability.* The source code of iPyCLES v1.0 and data and code needed to reproduce the figures in this paper are available from https://zenodo.org/doi/10.5281/zenodo.10911096 (Hu and Wright, 2024). iPyCLES v1.0 is released under the GPL-3.0 license. The
iPyCLES code includes Python, C, and Fortran code. Necessary dependencies include numpy, scipy, matplotlib, Cython, mpi4py, netCDF4, and C and Fortran compilers (the simulations presented in the paper were conducted using Python 3.7.10 and version 7.3.0 of the GNU compilers). The Fortran code is the radiation component, the Rapid Radiative Transfer Model (RRTMG), which is needed for the Isdac case: http://rtweb.aer.com/rrtm_frame.html. Code for the isotope computations is included in Isotopetracers.pyx, Csrc/isotope.h and Csrc/isotope-tracer.h. Code for the two-moment microphysics scheme is located in Microphysics_SB_2M.pyx, Csrc/microphysics_sb.h, and Crsc/micro-
physics_sb_ice.h. More details regarding iPyCLES v1.0 are available at: https://github.com/huzizhan/ipycles/tree/isotopetracer. MicroHH outputs are from simulations conducted using version 1.0 of the model (https://github.com/microhh/microhh) and were acquired from an archive published by Bastak Duran et al. (2021). Observational data from the BOMEX campaign were compiled by consulting the campaign report held by the University Corporation for Atmospheric Research (UCAR; https://www.eol.ucar.edu/sites/default/files/files_live/private/TD-9677_BOMEX_Dropsonde_Data.pdf). Observational data from the RICO campaign were acquired from the archive maintained by UCAR
(https://data.eol.ucar.edu/master_lists/generated/rico/). Observational data from the ISDAC campaign were acquired from the US Department of Energy Atmospheric Radiation Measurement Program data archive (https://www.arm.gov/research/campaigns/aaf2008isdac). Multi-model spreads and averages from model intercomparisons focused on the BOMEX and RICO campaigns were reconstructed manually from figures published by Siebesma et al. (2003) and van Zanten et al. (2011), respectively.

*Author contributions.* JSW and ZH conceived the overall framework of iPyCLES and selected the experiments. ZH carried out the model
development and evaluation. JSW and YP provided suggestions and advice on model development. JSW, ZH and MZ performed the data processing. All authors contributed to writing the manuscript.

*Competing interests.* The authors declare that they have no conflict of interest.

*Acknowledgements.* We thank Dr Yanluan Lin and Dr Xi Zhao for advice on implementing the microphysical parameterization and Dr Suqin Duan for advice on the isotope module. This research has been conducted with the support of the National Natural Science Foundation
of China (grant number 42275053), the Beijing Natural Science Foundation (grant number IS23121), and the Ministry of Science and Technology of the Peoples' Republic of China (2017YFC1501404).



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
