# Peer review of "iPyCLES v1.0: A New Isotope-Enabled Large-Eddy Simulator for Mixed-Phase Clouds"

_EGUsphere, 2024_

## Author Comment (AC1)

**Author Reply to RC1**

> This manuscript describes the implementation of a water isotopic tracer modeling capability plus a prognostic mixed phase microphysics scheme to an existing atmospheric large eddy simulation code, PyCLES, and presents results from three established intercomparison cases to assess the model performance. Given the potential of isotope based analyses for deepening understanding of the processes controlling the distribution of clouds and precipitation, the addition of another isotope enabled LES model is attractive. However, there are some key gaps in the formulation and evaluation of iPyCLES that need to be addressed.

We thank you for your careful reading of our manuscript and in-depth comments. In the following, we outline what we have done to address your concerns.

**Thermodynamics**

> RC1: I will begin with my greatest technical concern. The authors do not address the proper handling of PyCLES' specific entropy prognostic thermodynamic variable with the addition of the SB06 mixed phase microphysics scheme. In particular, they do not discuss how they deal with phase changes among water vapor, cloud ice, and cloud liquid occurring at non-equilibrium conditions (i.e., not subject to saturation adjustment). As Pressel et al 2015 discuss the nuances of employing a prognostic entropy equation in LES extensively and in particular as relates to phase partitioning, these authors should have been aware of the theoretical challenges in adding more complex microphysical schemes to PyCLES. However, this issue was not even alluded to, let alone addressed. The authors must explain how they adapt the thermodynamics of iPyCLES for compatibility with more general mixed-phase microphysics schemes. Given the temperature dependence of the fractionation processes, it would be reassuring to know that temperature is being diagnosed appropriately within the host atmospheric model.

We thank the reviewer for raising this issue, as it has highlighted an oversight in our implementation of the SB06 scheme that we will address in the revised submission. Specifically, although entropy sinks due to deposition were included, sinks due to ice formation processes other than deposition (e.g., immersion and contact freezing) were omitted.

For completeness: temperature is treated as a diagnosed variable in iPyCLES, calculated based on the prognostic variable specific entropy ($s$) following equations 29 and 30a-b of Pressel et al. (2015).

Prognostic updates to the thermodynamics are applied to $s$ following equation 7 of Pressel et al. (2015).

$$\frac{\partial s}{\partial t} + \frac{1}{\rho_0}\frac{\partial\left(\rho_0 u_i s\right)}{\partial x_i} = \frac{Q}{T} + \left(s_v - s_d\right)\left(\frac{dq_v}{dt}\right)_e - \frac{1}{\rho_0}\frac{\partial\left(\rho_0 \gamma_{s,i}\right)}{\partial x_i} + \dot{S}$$

The second term on the left-hand side is the momentum transport of $s$, the first term on the right-hand side is the source due to diabatic heating ($Q$), the second term on the right-hand side is the imposed vapor source tendency, the third term on the right-hand side collects the effects of subgrid turbulent mixing, and the final term is the irreversible entropy source, which includes non-equilibrium microphysical effects.

The mixed-phase saturation adjustment implementation from PyCLES still applies for the vapor-liquid equilibrium. We treat vapor-cloud droplet exchange using the standard saturation adjustment method as outlined in Appendix A of Pressel et al. (2015) but without ice; i.e., assuming $q_i$=0.
Specific entropy $s$ is altered due to latent heat release and temperature changes accordingly. The calculations are located in
https://github.com/huizizhan/ipycles/blob/isotopetracer/ThermodynamicsSB.pyx.
The difference from the original formulation is that the total water $q_t$ is replaced by $q_{vl}$ with $q_{vl} = q_t - q_i$.

Replacing the equilibrium assumption for ice-vapor exchange with the prognostic microphysics parameterization requires more substantial changes. The change in $s$ due to vapor-ice exchange is included in the irreversible entropy source ($\dot{S}$) along with other cloud microphysical processes.
Pressel et al. (2015) described the components of $\dot{S}$ as represented in PyCLES in their section 3.6.1.
To summarize, $\dot{S}$ includes

- Sink of entropy owing to the loss of mass when precipitation forms, referred to as $\dot{S}_P$ by Pressel et al. (2015).
- Source of entropy owing to the gain of water via evaporation, referred to as $\dot{S}_E$ by Pressel et al. (2015). (note that Pressel et al. (2015) omitted the dot in their equation 50)
- Entropy change due to water vapor diffusion into surrounding air through the saturated layer, referred to as $\dot{S}_D$ by Pressel et al. (2015). (as above, the dot was omitted in their equation 51)
- The entropy sink due to heating of hydrometeors falling into warmer air, either by maintaining raindrops at the wet-bulb temperature or by melting ice and snow below the freezing level, referred to as $\dot{S}_Q$ by Pressel et al. (2015).
- The entropy source to the surrounding air due to the aerodynamic drag of precipitation hydrometeors, as gravitational acceleration is balanced by the frictional dissipation of

kinetic energy in shear zones surrounding the hydrometeors, referred to as $\dot{S}_W$ by Pressel et al. (2015).

In our SB06 mixed-phase implementation, entropy sources and sinks related to the new representation of the ice phase are incorporated in $\dot{S}_P$. This treatment is conceptually similar to that adopted by Zhang et al. (2020) in implementing the Kaul et al. (2015) scheme in PyCLES. However, in responding to this comment we have realized that our implementation is incomplete, in that entropy sinks due to ice formation processes other than deposition are not considered. For the revised manuscript, we will revise the code to account for these entropy sinks and provide an explanation of all changes in our formulation relative to the framework outlined by Pressel et al. (2015).

The code for these calculations is located at
https://github.com/huzizhan/ipycles/blob/isotopetracer/Csrc/microphysics_sb_ice.h.

**The Blossey isotope model**

> RC1: In terms of presentation of the overall modeling framework, the authors are commendably thorough in documenting and referencing the equations and closures used for the isotopic module. The current approach appears to closely follow Blossey et al 2010 as regards the coupling to cloud microphysics. It would be extremely useful for the authors to provide a conceptual overview of their approach, noting important similarities and differences with prior approaches and highlighting any significant innovations. Additionally, discussion of the most significant uncertainties within the isotopic tracer module is warranted.

This is correct, our treatment of isotopic fractionation closely follows Blossey et al. (2010). The changes made so far are minor, and mainly toward adapting Blossey et al.'s approach to the specifics of the SB06 mixed-phase microphysics. For example, we have modified $b_l$ in equation B7 of Blossey et al. 2015 to fit the representation of the thermodynamics factor $G(T)$ implemented in the SB06 scheme. Our derivation of the revised form of $b_l$ can be found in the supplement to this reply. We are working on a schematic diagram to illustrate our approach.

Uncertainties in the isotopic parameterization include the following aspects, common to most if not all isotope-enabled cloud models:

- The choice of diffusivity ratios for heavier isotopes. For example, in these simulations we have adopted the diffusivities proposed by Cappa et al. (2003), which have been discussed quite critically in the literature, and there remains uncertainty as to the exact values of these parameters.

- Parameterized representations of non-equilibrium / kinetic fractionation processes. For example, the representation of kinetic fractionation during ice formation is designed for deposition and does not account for different ice nucleation processes, such as immersion and contact freezing of liquid droplets.

At this stage of the model development, we are mainly concerned with implementing passive tracers that behave like water isotopes rather than matching observations (see also our adoption of an initial profile based on Rayleigh distillation). We are currently designing cloud chamber experiments to test the various parameterizations of isotopic fractionation, and intend to use these experiments to revise and refine the current parameterizations for implementation in this model.

**Model valididition**

> In terms of model validation/evaluation, the authors make some efforts towards evaluating the results of their three case studies (notably, these case studies were implemented in the original PyCLES code and are not new to iPyCLES). However, the isotopic tracer predictions themselves are simply presented and discussed. In other words, there's no "ground truth" because none of the chosen test cases are accompanied by isotopic observations. There is also an absence of systematic sensitivity studies that could help to establish the internal consistency of the model and check for implementation bugs. In particular, the spikes around cloud top in the ISDAC test case are never satisfactorily explained; it's not even clear whether their occurrence is truly erroneous. Given the complexity of the isotopic coupling, there really needs to be more evidence presented that it has been correctly implemented.

We agree that the current state of validation for the iPyCLES model, and particularly its isotopic component, is incomplete. More experiments with different boundary conditions should be conducted and compared with "ground truth" observations. Here we have used the BOMEX, RICO, and ISDAC cases because we preferred to start with well-defined test cases that we knew PyCLES could simulate well. This way, problems could be more easily traced to our modifications, rather than to issues in implementing a new case. In addition, new cases would need to be introduced to a greater level of detail, lengthening an already long manuscript.
As mentioned above, our interests at this stage are more toward establishing that the isotopic tracers evolve according to expectation and identifying what additional information these tracers may provide relative to water alone. We will revise the manuscript text to make this point more clearly.

We are currently planning additional experiments focused on the recent MOSAiC and EUREC4A field campaigns to better evaluate the isotopic implementation in iPyCLES. We have multiple objectives for these experiments: (1) to validate and improve the current

implementation of the isotopic water cycle; (2) to clarify how isotopic observations can help to improve the model (not only in the isotopic component); (3) to evaluate how the model can aid the interpretation of isotopic observations; and (4) to identify new focus areas for model development to make best use of the isotopic tracers. Given the broad scope of these objectives, we believe that demonstrating the model first via cases that have been well tested and widely used is preferable to trying to do both at once.

As mentioned above, we also intend to use the model in conjunction with cloud chamber experiments, where environmental conditions can be controlled to mimic key processes. This includes isotopic equilibration of supercooled water in a sharp temperature gradient, which may explain the sharp peak in deuterium excess near the top of the mixed-phase cloud layer in the ISDAC case.

Although changes in deuterium excess are typically associated with kinetic fractionation processes, liquid-vapor equilibrium fractionation factors vary sharply in strong temperature gradients at low temperatures (Fig. R1).

[Figure]

**Figure R1.**

$d_v$

**Comparison with the Kaul15 scheme**

RC1: The comparison of the Kaul15 and SB06 schemes is interesting, but perhaps overstated, in that Kaul15 is not really intended as a high-fidelity mixed phase microphysics schemes in terms of capturing individual microphysical process rates. Also, the Kaul15 scheme could have been compared to the SBWarm/SB06 scheme for the RICO and BOMEX cases (including comparison of isotopes), but instead the authors chose to add a comparison to MicroHH (without any isotopic information).

We agree with your description of the Kaul15 parameterization. These limitations were our primary motivation for implementing the two-moment SB06 scheme to support isotopic processes. The point of the comparison is to establish the importance of the prognostic ice component with respect to the isotopic evolution.

Your point that the Kaul15 scheme could have been included in the evaluation of RICO and BOMEX is well taken. Results for the RICO case using the Kaul15 scheme are shown in Fig. R2 below. The simulation based on Kaul15 is bottom heavy compared to other simulations, with a weak anvil and rain water content distributed over a deeper layer. We will include information from the isotopic tracers based on this simulation in our revision.

[Figure]

**Figure R2.**

**Author Reply to RC2**

This is a well-written, and very interesting paper introducing a new isotope-enabled LES system to study cloud formation processes and their impact on the climate. The newly introduced isotope module is well described, and some first idealised test cases are presented. I missed a description or discussion of the potential to use the LES in more realistic meteorology setups and related to this the use of isotope observations available from recent campaigns such as EUREC4A for a more process-oriented evaluation of the modelling system. Also I am not entirely sure, why the authors present the BOMEX case,

> which I find a bit lengthy to read, but which (to me) brings not much interesting material, while the RICO and Arctic cases are super interesting. To some extent the summary and discussion section helps in this respect but I would have hoped for more guidance earlier in the text. Maybe help the reader understand why the BOMEX case is presented at all already in the introduction.

Thank you for your positive comments and your suggestions. We are planning to apply the model to the LES case linked to the EUREC4A campaign, and are interested in extending it to other cases with isotopic observations as well. Your points regarding the BOMEX case are well taken. We also considered whether to include this case before deciding that it would be helpful to demonstrate the isotopic tracers in this simpler case first. We plan to trim the description of this case and may remove it entirely in the revised manuscript.

> L.5: new sentence for the external forcings? I don't understand why they are mentioned together with isotopic fractionation, there's no direct link, right? And in a new sentence on external forcings: as I understood the available "facilities for external forcings" are idealised and don't allow for realistic meteorology setups, this should be discussed shortly at the end of the paper.

In the iPyCLES model, external forcing setups can be initialized based on realistic observational or reanalysis datasets as advective boundary conditions. Although the forcings do not involve fractionation, they are a means of introducing water vapor with different isotopic signatures. Here, we were considering the potential for distinguishing the role of moisture advection in feeding Arctic humidity inversions. We have begun to investigate this in sensitivity simulations, which will be presented in future work.

> L.17: "conspicuous peak of deuterium excess at the top of clouds" mention in which phase(s) you find this peak.

Thank you for this suggestion, we will specify that the peak is seen in water vapor.

> L. 44: It would help me if you could guide the reader a bit more for the following two paragraphs by starting with a sentence that explicitly states that you plan to use LES to address Morrison challenge 1 and isotopes to address challenge 2 and that you will expand on these two tools in the following two paragraphs.

Thank you for this excellent suggestion. We will adopt this approach in the revised manuscript.

> L. 72: "… a degree of dependence that depends on the extent of differential fractionation among the isotopes": I know what you mean, but I think this is not clear for non-isotope specialists. Please reformulate.

Thank you for pointing this out. We have rewritten it as "with the potential for additional information dependent on the extent to which each isotope is preferentially transferred during phase changes."

> L. 91: a recent publication on isotope effects and what we can learn from them on the dynamics and microphysics of tropical mixed phase clouds is de Vries et al. 2022 in addition to Bolot et al. 2013.

Thank you for this recommendation. We learned a lot from this paper and will cite it here in the revised submission.

> L. 101: Maybe Eckstein et al. 2018 is not the best reference for high-resolution simulations of cloud and boundary layer processes over short time scales. Since you also use shallow cumulus cases: Villiger et al. 2023 who performed COSMOiso simulations at 1 km horizontal grid spacing in the western tropical North Atlantic would be closer to what you mean here.

Great suggestion, thank you. We will change the citation in the revised submission.

> L. 105-108: Mentioning recent field campaigns with isotope observations, there's a key one that is missing here with observations of shallow cumulus clouds on 2 aircrafts, 4 ships and a land station. That's the EUREC4A campaign on elucidating the role of clouds-circulation coupling for climate.

You are absolutely correct and we are embarrassed to have overlooked it when writing this section. We will include it in the revised submission.

> L. 139: "...moisture and heat..." why heat?

> L. 147: "tightly intervowen with model thermodynamics" and dynamics?

Thank you for your careful reading. We will revise these as suggested.

> Eq. 1: not all terms are described.

Apologies for the oversight; we have added introductions to each term.

> L. 264: "Isotopes in rain and snow sediment according to the fall velocities computed for rain and snow in the microphysics scheme". Reformulate, it could be misunderstood, there's no fractionation in sedimentation, and Heavy and light water sediments with the same fall velocities.

We have rewritten this sentence as "Isotopes in rain and snow sediment without fractionation according to the fall velocities computed for rain and snow by the microphysics scheme"

> L. 282: I think the deuterium excess could be explained in a better way by mentioning that it is a tracer for non-equilibrium fractionation. At L. 280 it is not so clear what "mismatch" is exactly meant. The global meteoric water line is not introduced properly, so referring to it, while explaining the dexcess might not be too helpful for non-isotope readers.

> L. 283-284: a higher HDO concentration relative to H218O compared to equilibrium fractionation but not in absolute terms higher.

Thank you for pointing this out. We have changed 'mismatch' to 'divergence' as a callback to the previous sentence, and clarified that the higher HDO concentration implied by a larger deuterium excess is relative to equilibrium. We have also added a sentence to explicitly state that deuterium excess is a measure of the cumulative effects of non-equilibrium fractionation processes.

> L. 317: to what extent is this consistent with the Stewart 1978 approach? Pfahl et al. 2012 raise this very interesting and relevant question, which to my knowledge has never been addressed so far in the literature. An idealised modelling system such as iPyCLES could clarify this very elegantly.

The Lee and Fung (2008) approach is largely based on Stewart with some adjustments to account for different raindrop size distributions. We agree that iPyCLES can help to clarify the answer to this question, and are currently working with a nearby research institute to design cloud chamber experiments to evaluate it experimentally.

> L. 329: The diffusivites by Cappa et al. 2003 have been discussed quite critically in the literature, see Pfahl and Wernli, 2009. Maybe at least mention that there is some disagreement among experts about the best choice for the diffusivities.

Yes, we agree. This is an aspect of the model that we will treat more carefully as we work toward comparison with isotopic observations. We have noted the disagreement as suggested.

> L. 348: To me it was not clear at L. 290 that this was for SB06, so here I am a bit surprised and confused to what is done in which scheme.

Thank you for raising this concern, and apologies for the confusion. In the revised text, we have added clarifying remarks to indicate which processes apply to which microphyics parameterization.

> Eq. 33: interesting: in several recent publications (Aemisegger et al. 2021 and Villiger et al. 2022, Weather and Climate Dynamics), the influence of extratropical intrusions into the tropics have been highlighted, which are associated with enhanced slantwise subsidence from the extratropics. Could such a flow regime be represented with the simple idealised setup chosen in iPyCLES?

Among the modification we are looking to apply moving forward is a more robust two-dimensional model setup to investigate low-level regional overturning circulations, such as land-sea breeze systems. Such a setup would also be useful for looking at the influences of slantwise subsidence (there is, we think, some conceptual similarity to the simulations targeting the effects of moisture intrusions into the Arctic lower troposphere mentioned above).

> L. 390: I think the choice of the campaigns to evaluate the model is a bit surprising given that the EUREC4A effort would provide similar conditions as BOMEX and RICO but including isotope observations. But probably, this choice was due to the time frame of the project.

Indeed, this choice was dictated largely by the time frame of the project. We adopted the BOMEX and RICO cases because those two cases were already implemented and evaluated in the base model PyCLES, limiting uncertainties in step-by-step implementation and testing of the isotopic module. There is a prescription for LES simulations based on the EUREC4A campaign which we are very much looking forward to implementing in iPyCLES.

> L. 409: strictly speeking BOMEX happened in the tropics, isn't it more of a "trade wind cumulus test case"?

Changed as suggested.

> L445: I would expect a decrease of dexcess with height due to temperature decrease in a simple Rayleigh experiment (see e.g. Fig. A1 in Thurnherr et al. 2021 WCD, https://wcd.copernicus.org/articles/2/331/2021/wcd-2-331-2021.pdf)

We agree that our initial approach for vapor isotope initialization is unrealistic, and that this is something we must address as we move towards evaluating the model against isotopic observations. Thank you for the references on d-excess observations, which will be helpful for our future work.

> Analyses related to Fig. 8: how different are cloudy vs. non-cloudy profiles vs. dry-warm (subsidence dominated) profiles? Villiger et al. 2023 proposed a simple way of evaluating isotope signals from numerical models in their representation of shallow cumulus clouds. Are the differences between cloudy and non-cloudy profiles similar as in their study? I very much recomment the authors to do this analysis because it allows them to relate their simulations to observations from EUREC4A in an elegant and simple way.

Thank you very much for this suggestion. Although we did not generate these outputs, we are running the simulations to generate them now and plan to include this analysis in the revised submission.

> L. 500-510: this is a very interesting paragraph. How comes that the evaporation of hydrometeors is maximum in the cloud layer. I struggle a bit with understanding that. Is the cloud layer undersaturated? Mainly in downdrafts? Or Is this evaporation from anvil precipitation from above falling into subsaturated layers? Can this analysis be done for cloudy and non-cloudy profiles?

See previous response. Based on the cloud fraction and cloud water content profiles, we suspect that it is from anvil precipitation falling into subsaturated layers. iPyCLES produces a relatively pronounced anvil relative to other models (see Fig. R2 above).

> L 516: surprising that rain is slightly more enriched than cloud liquid water given that the rain most likely originates from above and cloud water might be of more local origin (?), so is this again a signature of rain evaporation?

Yes, we believe that this is a signature of partial evaporation and incomplete equilibration of the raindrops. It may also reflect spatial heterogeneities in the vapor composition between the cloud core and anvil cells. We plan to evaluate this in more detail after generating the outputs for cloudy and non-cloudy profiles.

> Fig. 8 and on the previous point: I would find a disequilibrium analysis quite useful here, because it is relatively cumbersome to compare the delta vapour to the delta liquid and rain water content to know which phase is enriched compared to the other (see de Vries et al. 2022 ACP, https://acp.copernicus.org/articles/22/8863/2022/acp-22-8863-2022.html)

Thank you for this good suggestion. We will adapt the figures to use this approach.

> L. 655: these measurements are available from EUREC4A.

Indeed. We have revised this sentence to include this information and our intention to conduct the EUREC4A LES intercomparison simulation in follow-up work.

> To me the Arctic mixed phase case is very much focused on the two microphysical schemes and their comparison. I find that the isotope aspects come a bit more as a byproduct and I find the dexcess peak at cloud top very suspicious.

We agree that the sharpness of the peak is suspicious, but we think that this signal is worth exploring because of the complexity of the microphysical processes, thermodynamic structure, and turbulent mixing there. Although we do not have a full answer yet for why this signature shows up in the SB06 simulation, the very sharp vertical variations in fractionation factors (Fig. R1 above) imply that isotopic exchange between supercooled water and water vapor occurring in such sharp temperature gradients would be both interesting and informative.

> 725-727: yes, I totally agree on the need for combined use of observations and high-resolution model simulations to be able to fully use the potential of isotopes as tracers. And I

> really think iPyCLES makes a significant contribution to this. But: then can you give more details about how you would recommend to run the system in realistic meteorology setups to make the simulations comparable to observations?

Thank you for this question. We currently envision iPyCLES mainly as an idealized testbed for designing experiments to test hypotheses developed from individual field campaigns. It can also serve as a useful addition to LES model intercomparisons based on individual field campaigns, to help to better engage the information in isotopic observations. Currently the applicability of the model to real cases is limited by the simplicity of the surface boundary conditions, which is a target for development in the next stage of this project. We have added text in the revised manuscript to better communicate these strengths and limitations.

> L. 180: "smaller **than** the freezing point..."

> L. 197: dec**r**ibed

> L. 280 ff: d a variable and should be in italics, while 0 is the oxygen atom and should be in normal font.

> L. 386: replace "conditionsis" by "conditions".

> L. 662: "of of" remove one "of"

Thank you for identifying these typos and mistakes, which have been corrected in the revised manuscript.

**Isotopic exchange during rain evaporation**

In iPyCLES, the thermodynamic factor for isotopic exchange during rain evaporation $G(T)^{\mathrm{iso}}$ is parameterized following Blossey et al. (2010). However, the cloud microphysics parameterization used by Blossey et al. (2010) is a one-moment scheme developed by Lin et al. (1983), which differs from Seifert and Beheng (2006, hereafter SB06) in terms of vapor exchange during rain evaporation. The water mass loss in rain evaporation in the parameterization used by Blossey et al. (2010) is represented as:

$$\frac{dm}{dt} = -2\pi D_r F_v D_v \frac{1 - S_l}{1 + b_l} \rho_v^* \tag{1}$$

where $m$ is the mass of raindrops, $D_r$ is the diameter of raindrop, $F_v$ is the ventilation factor, $D_v$ is the vapor diffusivity, $\rho_v^*$ is the density of saturated air, and $S$ is the saturation ratio $S = p_v / p_v^*$. The parameter $b_l$ is defined as

$$b_l \equiv \frac{D_v L_v^2 \rho_v^*}{\kappa R_v T^2} \tag{2}$$

where $L_v$ is the latent heat of vaporization, $\kappa$ is the thermal conductivity, $T$ is temperature, and $R_v$ is the gas constant of water vapor.

The isotope parameterization is developed based on the formulation above, using the variable $b_l$:

$$\frac{dm^{\mathrm{iso}}}{dt} = -2\pi D_r F_v^{\mathrm{iso}} D_v^{\mathrm{iso}} \left[ \frac{R_{rain}}{\alpha_{lv}} \times \left( \frac{1 + b_l S}{1 + b_l} \right) - R_{vapor} \times S \right] \rho_v^* \tag{3}$$

where $R_{rain}$ and $R_{vapor}$ are the isotopic ratios in rain and water vapor and $\alpha_{lv}$ is the fractionation factor of liquid-vapor equilibrium fractionation.

The expression of $\frac{dm}{dt}$ during rain evaporation in the SB06 scheme used in our model follows Pruppacher and Klett (1997):

$$\frac{dm}{dt} = 2\pi D_r F_v (S - 1) G(T) \tag{4}$$

where $G(T)$ is given by:

$$G(T) = \left[ \frac{R_v T}{D_v p_v^*} + \left( \frac{L_v}{R_v T} - 1 \right) \left( \frac{L_v}{\kappa T} \right) \right]^{-1} \tag{5}$$

Therefore, modifications must be made to adapt the isotopic loss based on equation (3) to the revised formulation of rain evaporation in equation (4). Here, we derive the appropriate form for a new parameter $b_l'$ to replace $b_l$ and implement isotopic exchange in the SB06 parameterization.

**Derivation of $b_l$**

The vapor loss during rain evaporation is modeled based on the vapor density gradient subject to a prescribed diffusivity:

$$\frac{dm}{dt} = -2\pi D_r F_v D_v \left[ \rho_v(s) - \rho_v(\infty) \right] \tag{6}$$

where $\rho_v(s)$ is the vapor density at the raindrop surface and $\rho_v(\infty)$ is the vapor density in the ambient environment. Heat added to the drop is expressed as:

$$Q = -2\pi D_r F_v \kappa [T(s) - T(\infty)] \tag{7}$$

which is balanced by latent heat release, calculated as the product of the rate of change in raindrop mass and the latent heat of vaporization $L_v$, i.e.:

$$\frac{dm}{dt} L_v + Q = 0 \tag{8}$$

The temperature and vapor density gradients are related by:

$$\kappa [T(s) - T(\infty)] = -D_v [\rho_v(s) - \rho_v(\infty)] L_v \tag{9}$$

The relationship between saturation vapor density and temperature is given by the Clausius–Clapeyron relation:

$$\frac{d\rho_v^*}{dT} \approx \frac{\rho_v^* L_v}{R_v T^2} \tag{10}$$

We apply $T(s)$ and $T(\infty)$ in equation (10) and take the exponential of both sides, yielding:

$$\frac{\rho_v^*[T(s)]}{\rho_v^*[T(\infty)]} = \exp \frac{L_v}{R_v} \left[ \frac{1}{T(\infty)} - \frac{1}{T(s)} \right] \tag{11}$$

Applying Taylor expansion yields

$$\frac{\rho_v^*[T(s)]}{\rho_v^*[T(\infty)]} \approx 1 + \frac{L_v}{R_v} \left[ \frac{1}{T(\infty)} - \frac{1}{T(s)} \right] \tag{12}$$

With the approximation $T(\infty)T(s) \approx T^2$, we have:

$$\frac{L_v}{R_v} \left[ \frac{1}{T(\infty)} - \frac{1}{T(s)} \right] \rightarrow \frac{L_v}{R_v T^2} [T(s) - T(\infty)] \rightarrow \frac{L_v \Delta T}{R_v T^2} \tag{13}$$

Equation (9) can then be approximated as:

$$\rho_v^*[T(s)] \approx \rho_v^*[T(\infty)] \times \left( 1 + \frac{L_v \Delta T}{R_v T^2} \right) \tag{14}$$

The vapor density at the raindrop surface ($\rho_v(s)$) can be approximated as the saturation vapor density at temperature $T(s)$, so that $\rho_v(s) - \rho_v(\infty)$ can be rewritten as:

$$\rho_v(s) - \rho_v(\infty) = \rho_v^*[T(\infty)] \left( 1 + \frac{L_v \Delta T}{R_v T^2} \right) - \rho_v(\infty) \tag{15}$$

$$= \left[ 1 + \frac{L_v \Delta T}{R_v T^2} - S \right] \rho_v^*[T(\infty)] \tag{16}$$

Subsituting equation (15) into equation 6, we have:

$$\frac{dm}{dt} = -2\pi D_r F_v D_v \left[1 + \frac{L_v \Delta T}{R_v T^2} - S\right] \rho_v^*[T(\infty)] \tag{17}$$

and the mass-heat balance equation (eq. 9) can be rewritten as:

$$\kappa \Delta T = -D_v \left[1 + \frac{L_v \Delta T}{R_v T^2} - S\right] \rho_v^*[T(\infty)] L_v \tag{18}$$

The temperature difference $\Delta T$ is given by:

$$\Delta T = -\frac{D_v (1-S) \rho_v^* L_v}{\kappa + D_v \frac{L_v^2}{R_v T^2} \rho_v^*} \tag{19}$$

where $\rho_v^*$ refers to $\rho_v^*[T(\infty)]$ calculated using the ambient temperature $T(\infty)$.

Bringing this expression for $\Delta T$ into equation (17) yields:

$$\frac{dm}{dt} = -2\pi D_r F_v D_v \left[1 + \frac{L_v}{R_v T^2} \left(-\frac{D_v (1-S) \rho_v^* L_v}{\kappa + D_v \frac{L_v^2}{R_v T^2} \rho_v^*}\right) - S\right] \rho_v^* \tag{20}$$

After derivation, we have:

$$\frac{dm}{dt} = -2\pi D_r F_v D_v \left[(1-S)(1 - \left(\frac{1}{\frac{\kappa R_v T^2}{D_v \rho_v^* L_v^2} + 1}\right))\right] \rho_v^* \tag{21}$$

Taking $b_l$ as:

$$b_l \equiv \frac{D_v L_v^2 \rho_v^*}{\kappa R_v T^2} \tag{22}$$

and using the transformation $1 - \frac{1}{a+1} = \frac{1}{1+\frac{1}{a}}$, we have:

$$\frac{dm}{dt} = -2\pi D_r F_v D_v \frac{1-S}{1+b_l} \rho_v^* \tag{23}$$

**New formulation of $b_l'$**

Unlike the vapor density difference approximation applied above, the parameterization used in the SB06 scheme is based on the assumption of differences in vapor density in a saturated water vapor field are related to temperature differences:

$$\frac{d\rho_v}{\rho_v} = \frac{L_v}{R_v T^2} dT - \frac{1}{T} dT \tag{24}$$

To solve this differential equation, we take the integral from $T(s)$ to $T(\infty)$:

$$\ln \rho_s[T(s)] - \ln \rho[T(\infty)] = -\frac{L_v}{R_v T(s)} - \ln T(s) + C - (-\frac{L_v}{R_v T(\infty)} - \ln T(\infty) + C) \tag{25}$$

Again adopting the approximation $T(\infty)T(s)=T^2$, we have

$$\ln \frac{\rho_s[T(s)]}{\rho_s[T(\infty)]} = \frac{L_v}{R_v T(\infty)} - \frac{L_v}{R_v T(s)} - (\ln T(s) - \ln T(\infty)) \tag{26}$$

$$= \frac{L_v}{R_v T^2}[T(s) - T(\infty)] - \ln \frac{T(s)}{T(\infty)} \tag{27}$$

where $\frac{T_r}{T_\infty} \approx 1$ so $\ln \frac{T_r}{T_\infty} \approx 0$. Taking the exponential of both sides, we have:

$$\frac{\rho_v^*(T(s))}{\rho_v^*(T(\infty))} \approx \exp \frac{L_v}{R_v T^2}[T(s) - T(\infty)] \tag{28}$$

Applying Taylor expansion leads to

$$\frac{\rho_v^*[T(s)]}{\rho_v^*[T(\infty)]} \approx 1 + \frac{\Delta T}{T}\left(\frac{L_v}{R_v T} - 1\right) \tag{29}$$

$$\rho_v^*[T(s)] \approx \rho_v^*[T(\infty)]\left[1 + \frac{\Delta T}{T}\left(\frac{L_v}{R_v T} - 1\right)\right] \tag{30}$$

Taking equation (30) into equation (15) with the same approximation $\rho_v(s) = \rho_v^*[T(s)]$:

$$\rho_v(s) - \rho_v(\infty) = \left[1 + \frac{\Delta T}{T}\left(\frac{L_v}{R_v T} - 1\right) - S\right]\rho_v^*[T(\infty)] \tag{31}$$

Substituting this equation into equation (9) yields:

$$\kappa \Delta T = -D_v\left[1 + \frac{\Delta T}{T}\left(\frac{L_v}{R_v T} - 1\right) - S\right]\rho_v^*[T(\infty)]L_v \tag{32}$$

$\Delta T$ can be rewritten as

$$\Delta T = -\frac{D_v \rho_v^*(1 - S_l)L_v}{\kappa + D_v \rho_v^* \frac{L_v}{T}\left(\frac{L_v}{R_v T} - 1\right)} \tag{33}$$

with $\rho_v^* = \rho_v^*[T(\infty)]$. By bringing equation (17) and equation (33) into $\frac{dm}{dt}$, we have

$$\frac{dm}{dt} = -2\pi D_r F_v D_v\left[1 + \frac{d_l}{T}\left(-\frac{D_v \rho_v^*(1 - S)L_v}{\kappa + D_v \rho_v^* \frac{L_v}{T}d_l}\right) - S\right]\rho_v^* \tag{34}$$

where

$$d_l \equiv \left(\frac{L_v}{R_v T} - 1\right) \tag{35}$$

Simplifying equation (34)

$$\frac{dm}{dt} = -2\pi D_r F_v D_v\left[(1 - S) - (1 - S)\frac{d_l}{T}\left(\frac{1}{\frac{\kappa}{D_v \rho_v^* L_v} + \frac{D_v \rho_v^* \frac{L_v}{T}d_l}{D_v \rho_v^* L_v}}\right)\right]\rho_v^* \tag{36}$$

$$= -2\pi D_r F_v D_v\left[(1 - S) - (1 - S)\left(\frac{1}{\frac{\kappa T}{D_v \rho_v^* L_v d_l} + 1}\right)\right]\rho_v^* \tag{37}$$

$$= -2\pi D_r F_v D_v\left[(1 - S)\left(1 - \frac{1}{\frac{\kappa T}{D_v \rho_v^* L_v d_l} + 1}\right)\right]\rho_v^* \tag{38}$$

Applying $1 - \frac{1}{a+1} = \frac{1}{1+\frac{1}{a}}$, in the end we have

$$\frac{dm}{dt} = -2\pi D_r F_v D_v \frac{1-S}{1+b_l'} \rho_v^*$$

(39)

with $b_l'$ as

$$b_l' \equiv \frac{D_v \rho_v^* L_v d_l}{\kappa T} = D_v \rho_v^* \frac{L_v}{\kappa T} \left( \frac{L_v}{R_v T} - 1 \right)$$

To test the derivation above, we take the $b_l'$ back into equation (39)

$$\frac{dm}{dt} = 2\pi D_r F_v D_v \rho_v^* \frac{S-1}{1 + D_v \rho_v^* \frac{L_v}{\kappa T} \left( \frac{L_v}{R_v T} - 1 \right)}$$

(40)

$$\frac{dm}{dt} = 2\pi D_r F_v (S-1) \frac{D_v \rho_v^*}{1 + D_v \rho_v^* \frac{L_v}{\kappa T} \left( \frac{L_v}{R_v T} - 1 \right)}$$

(41)

$$\frac{dm}{dt} = 2\pi D_r F_v (S-1) \left[ \frac{1}{D_v \rho_v^*} + \left( \frac{L_v}{R_v T} - 1 \right) \left( \frac{L_v}{\kappa T} \right) \right]^{-1}$$

(42)

where, applying the ideal gas law:

$$\frac{1}{D_v \rho_v^*} = \frac{R_v T}{D_v p_v^*}$$

(43)

Equation (39) can thus be applied in equation (4) with the thermodynamic factor $G(T)$, confirming that the new expression of $b_l$ can be used to adapt the isotopic parameterization of Blossey et al. (2010) to the SB06 microphysical scheme.

**References**

Blossey, P. N., Kuang, Z., and Romps, D. M.: Isotopic Composition of Water in the Tropical Tropopause Layer in Cloud-Resolving Simulations of an Idealized Tropical Circulation, Journal of Geophysical Research: Atmospheres, 115, https://doi.org/10.1029/2010JD014554, 2010.

Lin, Y.-L., Farley, R. D., and Orville, H. D.: Bulk Parameterization of the Snow Field in a Cloud Model, Journal of Applied Meteorology and Climatology, 22, 1065–1092, https://doi.org/10.1175/1520-0450(1983)022¡1065:BPOTSF¿2.0.CO;2, 1983.

Pruppacher, H. R. and Klett, J. D.: Microphysics of clouds and precipitation, vol. 18, Dordrecht: Kluwer Academic Publishers, https://doi.org/10.1007/978-0-306-48100-0, 1997.

Seifert, A. and Beheng, K. D.: A Two-Moment Cloud Microphysics Parameterization for Mixed-Phase Clouds. Part 2: Maritime vs. Continental Deep Convective Storms, Meteorology and Atmospheric Physics, 92, 67–82, https://doi.org/10.1007/s00703-005-0113-3, 2006.